# Proteomic Studies of Primary Acute Myeloid Leukemia Cells Derived from Patients Before and during Disease-Stabilizing Treatment Based on All-Trans Retinoic Acid and Valproic Acid

**DOI:** 10.3390/cancers13092143

**Published:** 2021-04-29

**Authors:** Maria Hernandez-Valladares, Rebecca Wangen, Elise Aasebø, Håkon Reikvam, Frode S. Berven, Frode Selheim, Øystein Bruserud

**Affiliations:** 1Department of Clinical Science, University of Bergen, 5021 Bergen, Norway; rebecca.wangen@uib.no (R.W.); elise.aasebo@uib.no (E.A.); hakon.reikvam@uib.no (H.R.); 2The Proteomics Facility of the University of Bergen (PROBE), University of Bergen, 5009 Bergen, Norway; frode.berven@uib.no (F.S.B.); frode.selheim@uib.no (F.S.); 3Department of Medicine, Haukeland University Hospital, 5021 Bergen, Norway; 4The Department of Biomedicine, University of Bergen, 5009 Bergen, Norway

**Keywords:** acute myeloid leukemia, treatment, chemosensitivity, all-trans retinoic acid, valproic acid, histone deacetylase, mass spectrometry, proteomics, phosphoproteomics

## Abstract

**Simple Summary:**

Acute myeloid leukemia (AML) is an aggressive hematological malignancy, and the only possibility of cure is intensive antileukemic treatment. Such treatment is only possible for relatively young and fit patients, but there is a large subset of elderly/unfit patients above 65–70 years of age who cannot receive intensive treatment. The low-toxicity combination of valproic acid (VP) plus all-trans retinoic acid (ATRA) has an AML-stabilizing effect for a subset of patients, even those with chemoresistant relapse. This response may be an indirect effect due to the modulation of the metabolic environment, but in this article, we describe that the in vivo treatment with ATRA/VP also has direct effects on the fundamental functions of human AML cells. Furthermore, new and less intensive antileukemic therapies are now available (i.e., venetoclax, azacitidine, decitabine), and the combination of these new agents with ATRA/VP may represent a new therapeutic strategy in AML.

**Abstract:**

All-trans retinoic acid (ATRA) and valproic acid (VP) have been tried in the treatment of non-promyelocytic variants of acute myeloid leukemia (AML). Non-randomized studies suggest that the two drugs can stabilize AML and improve normal peripheral blood cell counts. In this context, we used a proteomic/phosphoproteomic strategy to investigate the in vivo effects of ATRA/VP on human AML cells. Before starting the combined treatment, AML responders showed increased levels of several proteins, especially those involved in neutrophil degranulation/differentiation, M phase regulation and the interconversion of nucleotide di- and triphosphates (i.e., DNA synthesis and binding). Several among the differentially regulated phosphorylation sites reflected differences in the regulation of RNA metabolism and apoptotic events at the same time point. These effects were mainly caused by increased cyclin dependent kinase 1 and 2 (CDK1/2), LIM domain kinase 1 and 2 (LIMK1/2), mitogen-activated protein kinase 7 (MAPK7) and protein kinase C delta (PRKCD) activity in responder cells. An extensive effect of in vivo treatment with ATRA/VP was the altered level and phosphorylation of proteins involved in the regulation of transcription/translation/RNA metabolism, especially in non-responders, but the regulation of cell metabolism, immune system and cytoskeletal functions were also affected. Our analysis of serial samples during the first week of treatment suggest that proteomic and phosphoproteomic profiling can be used for the early identification of responders to ATRA/VP-based treatment.

## 1. Introduction

Acute myeloid leukemia (AML) is an aggressive hematological malignancy and the median age of the first time of diagnosis is 65–70 years [1]. The acute promyelocytic leukemia variant should be regarded as a separate entity characterized by specific genetic abnormalities, different treatment, and better prognosis than patients with the non-promyelocytic variants [2], and when we use the term AML in our present article, this refers to the non-promyelocytic variants. The only possibility to cure AML is intensive chemotherapy, possibly in combination with autologous or allogeneic stem cell transplantation [1]. However, intensive therapeutic strategies are usually possible only for younger patients (i.e., below 70–75 years of age) without severe comorbidity, whereas the large group of elderly and/or unfit patients can only receive less intensive and thereby only AML-stabilizing treatment due to an unacceptable risk of severe treatment-related morbidity or mortality [1]. Such AML-stabilizing treatment has previously been based on low-dose and thereby low-toxicity therapy with conventional cytotoxic drugs (e.g., melphalan, cytarabine, 6-mercaptopurine, hydroxyurea) or the demethylating agents azacitidine and decitabine [3,4,5,6,7]. Recent studies suggest that combinations of demethylating agents with the BCL2 apoptosis regulator (BCL2) antagonist venetoclax is very effective and well tolerated in AML; this stabilizing treatment can induce complete remission for a larger fraction of patients and may also be effective in patients with chemoresistant AML relapse [8,9]. Finally, antileukemic treatment with low-toxicity cytotoxic drugs (i.e., low-dose cytarabine, 6-mercaptopurine or hydroxyurea) in combination with all-trans retinoic acid (ATRA) plus valproic acid (VP) can also induce disease stabilization for 30% of patients and may even induce complete remissions [4,5,6]. This last strategy based on the ATRA/VP combination has a very low toxicity; it may be effective in patients with AML relapse as well [4,6] and its effect has been documented in several clinical studies [5]. The use of ATRA in combination with decitabine was even associated with prolonged survival in a recent clinical study [10]. However, it is not known whether these two drugs can increase the efficiency of venetoclax.

The antileukemic effects of ATRA and VP have been investigated in both experimental and clinical studies. The retinoic acid receptors (RARs) and retinoid X receptors (RXRs) are ligand-activated nuclear receptors [11,12]. ATRA is a potent vitamin A derivative and a high affinity activating ligand of the three RARα/β/γ receptors; it can thereby be involved in the transcriptional regulation of hundreds of genes [13]. Several non-retinoid compounds including fatty acids are also potent RXR activators [14,15], but it is not known whether ATRA therapy might have additional indirect effects on AML cells caused by its modulation of the systematic levels of several lipid metabolites [16]. Finally, clinical studies also suggest that the antileukemic effects of ATRA could be detected, especially for patients with an altered expression of chromatin-modifying genes, MDS1 and EVI1 complex locus (*MECOM*) overexpression, nucleophosmin 1 (*NPM1*) mutations or isocitrate dehydrogenase (*IDH*) mutations with lysine demethylase 1A (KDM1A) deregulation [13,17].

Protein acetylation is regulated by the balance between histone acetyltransferases (HATs) and histone deacetylases (HDACs) [18]. The HDAC enzymes are grouped into Class I (HDAC 1/2/3/8), class II (HDAC 4/5/6/7/9/10), class III (the sirtuins SIRT1-7) and Class IV (only HDAC11) [18]. HDAC inhibitors alter the acetylation of several proteins, including histones, causing increased gene transcription [19,20]. VP is a branched short-chain fatty acid that targets two of the four classes of HDACs: Class I, subclasses Ia and Ib; and Class II, subclass IIa [21]. However, the exceptions are HDAC 9/11, which are activated and HDAC 6/8/10, that do not seem to be affected by VP [18,21,22]. Several other HDAC inhibitors are also in clinical trials; the inhibitory profile varies between individual inhibitors but the inhibition of Class I and Class II HDACs is common both for VP, vorinostat, belinostat and panobinostat [18]. VP seems to have antiproliferative and proapoptotic in vitro effects on primary human AML cells, although these effects differ between patients [6,23]. VP also alters the systemic metabolic regulation of especially lipid (including fatty acid) and amino acid metabolism [24]. These effects are similar to those observed with ATRA and may thus contribute to the antileukemic activity of the combined treatment [14,15]. Finally, retinoic acid receptors can influence the regulation of gene expression by altering the balance between HDACs and HATs [13]. Taken together, these observations suggest that the combination of RARA/RXRA agonists and HDAC antagonists should be further explored in human AML, possibly as a part of combination therapy also including other targeted therapies, e.g., demethylating agents, KDM1A inhibitors, MECOM inhibitors, cytarabine, CD38 molecule (CD38) targeting or even BCL2 antagonists [13].

As described above, both ATRA and VP have antileukemic effects and have been tried as a low-toxicity AML alternative of stabilizing treatment. These two pharmacological agents may not represent the final solution for the clinical strategies of RARA and HDAC inhibition, respectively. However, detailed studies of the in vivo effects of the two agents on primary AML cells may help us understand the mechanisms behind the antileukemic effects of these two therapeutic strategies. Such information will probably also be relevant if more potent rexinoids and/or alternative HDAC inhibitors become available for clinical use [20,25].

In this context, we therefore used mass spectrometry (MS)-based proteomic and phosphoproteomic strategies to (i) compare the primary AML cells of responders and non-responders to ATRA/VP and (ii) to investigate the effects of in vivo treatment with ATRA/VP on primary AML cells. 

## 2. Materials and Methods

### 2.1. The Treatment Protocols Based on ATRA, VP and Low-Dose Cytotoxic Therapy

The patients were included in two nonrandomized Phase II clinical studies (see the Institutional Review Board Statement at the end of this manuscript), which were based on combined therapy with ATRA (22.5 mg/m^2^ orally twice daily for 14 days every 12th week), continuous oral VP treatment at the maximum tolerated doses and low-dose therapy with hydroxyurea, 5-mercaptopurine and/or cytarabine [4,6].

The first study (REK Vest 215.03; also referred to as the ATRA–VP–TP (theophylline) study in this manuscript [6]) included 24 patients, nine of which were classified as responders. Patients in this study received ATRA from day 1 as described above with 12 weeks intervals (Figure 1). The other drugs were administered according to the following guidelines:VP. On day three of the first cycle, the patients received intravenous VP first as a loading dose (5 mg/kg for 60 min) and thereafter as a continuous infusion (28 mg/kg/24 h) until day 8; the patients thereafter received continuous oral treatment with the highest tolerated dose and the target serum level being 300–600 μmol/L.TP. The patients received intravenous TP on day 3 with a loading dose (5 mg/kg) and thereafter continuous infusion (0.65 mg/kg/hour) until day 8; they later received continuous oral therapy and the target serum level was 50–100 μM.Cytotoxic drugs. Patients with circulating AML blasts > 50 × 10^9^/L at the time of diagnosis or later increasing circulating leukemic blasts received cytotoxic drugs to achieve stable AML blast levels below 50 × 10^9^/L.

The second study (REK Vest 231.06; referred to as the ATRA–VP–AraC (cytarabine) study [4]) included 36 patients, among which 11 were classified as responders. These patients received VP monotherapy for the first seven days with loading dose, 24 h of intravenous infusion as described above, and thereafter oral VP treatment (target level 300–600 μmol/L (Figure 1). On days 8–22, the patients received oral ATRA, and on days 15–24, they received subcutaneous AraC 10 mg/m^2^ once daily. ATRA and AraC were repeated with 12 weeks intervals. If the peripheral blood blasts levels increased later (>50 × 10^9^/L), the patients received either hydroxyurea or 5-mercaptopurine instead of AraC. 

### 2.2. Preparation of Enriched AML Cells

Our strategy for the selection of patients with circulating AML cells and standardized preparation of enriched AML cells has been described and discussed in detail previously [26,27,28]. Circulating AML cells were collected from 28 patients with >80% AML blasts among circulating leukocytes before the start of treatment (day 1). Five responders and six non-responders were included from the ATRA–VP–TP study whereas six responders and eleven non-responders were included from the ATRA–VP-AraC study. The clinical and biological characteristics of each patient are presented in Table 1, together with the peripheral blood leukocyte counts before the start of treatment. These pre-treatment cell samples are referred to as the responder (PRE-R) or non-responders (PRE-NR) patient/samples, according to the final clinical response of the patient (Figure 1). 

Patient cell samples from 10 patients were also available from the ATRA–VP–TP study at day 3 (samples treated with ATRA from day 1) and at day 8 (samples treated with ATRA from day 1 and additional VP and TP from day 3). These patient/samples were classified as responder and non-responder as described above: five responders at day 3 (referred to as 3D-R), five non-responders at day 3 (3D-NR), five responders at day 8 (8D-R) and five non-responders at day 8 (8D-NR) (Figure 1).

Blood samples were collected on ACD tubes. Due to the relatively high levels of circulating AML cells, enriched leukemia cell populations (generally >95%) could be prepared by density gradient separation alone (Lymphoprep, Axis-Shield; specific density 1.077) [29]. Our patients thus represent a group of consecutive patients with high peripheral blood blast levels (i.e., >80% of circulating leukocytes being blast cells, leukocytes >20 × 10^9^/L); the blast count then being the only criteria for patient selection. Membrane molecule expression and karyotype were analyzed as a part of the routine clinical handling of these patients. The methods for molecular genetic analyses have been described in a previous publication [30].

### 2.3. Patient Sample Preparation for MS-Based Proteomics and Phosphoproteomics Analysis

Patient cell samples were lysed in 4% sodium dodecyl sulfate (SDS)/0.1 M Tris-HCl (pH 7.6) and immobilized metal affinity chromatography (IMAC) has been described elsewhere [31]. Briefly, 20 µg of each patient lysate was mixed with 10 µg of the AML super-SILAC (stable isotope labeling by amino acids in cell culture) mix [32] for proteomic analyses, and processed according to the filter-aided sample preparation (FASP) protocol [31,33]. The super-SILAC spiked peptide samples were further fractionated using the Pierce high pH-reversed-phase peptide fractionation kit (Thermo Fisher Scientific, Bartlett, IL, USA). Three fractions containing peptides eluted with 10%, 17.5% and 50% acetonitrile (ACN) were collected. The phosphoproteomics samples (128–1432 µg range) were mixed with the super-SILAC mix in an 1:2 ratio (w:w; super-SILAC mix:AML patient sample), FASP processed and enriched for phosphopeptides using the IMAC procedure.

### 2.4. LC–MS/MS Measurements

Peptide sample preparation prior to liquid chromatography with tandem mass spectrometry (LC–MS/MS) and settings of the LC–MS/MS runs on a Q Exactive HF Orbitrap mass spectrometer coupled to an Ultimate 3000 Rapid Separation LC system (Thermo Scientific, Waltham, MA, USA) were conducted as described earlier for global proteomics and phosphoproteomics samples [26], with the exception of the 195 min elution gradients used in the LC settings to analyze the first and the third proteome fractions. The first of them was eluted with a gradient starting at 5% B (0.1% formic acid/ACN) from 0 to 5 min, and increased to 14% B from 5 to 90 min, then to 32% B from 90 to 140 min and to 90% B from 140 to 155 min. Then, it was held at 90% B from 155 to 170 min, then ramped to 5% B from 170–175 min and held at 5% B for the last 20 min. The third proteome fraction was eluted with a gradient starting at 5% B from 0 to 5 min and increased to 18% B from 5 to 40 min, then to 38% B from 40 to 140 min and to 90% B from 140 to 155 min. The last 40 minutes of the run were as described for the first fraction.

The super-SILAC proteomic and phosphoproteomic samples were analyzed separately in a controlled randomized order. We used a HeLa protein digest run for approximately every 9th or 12th patient samples as LC–MS/MS quality controls. 

### 2.5. Data and Bioinformatics Analysis

LC–MS/MS raw files from 11 responder and 17 non-responder samples from both studies before treatment at day 1 (i.e., PRE-R and PRE-NR), and from 5 responder and 5 non-responder samples (from the ATRA–VP–TP study only) at day 1, day 3 and day 8 (i.e., PRE-R, PRE-NR, 3D-R, 3D-NR, 8D-R and 8D-NR) were processed with MaxQuant software version 1.5.2.8 separately [34,35]. MaxQuant search parameters are described elsewhere [26]. The spectra were analyzed using the concatenated forward and reversed-decoy Swiss-Prot Homo sapiens database version 2018_02 using the Andromeda search engine [36]. The Perseus 1.6.14.0 platform was used to analyze the protein groups and phosphosites MaxQuant-generated output files [37]. MaxQuant-normalized SILAC ratios were treated as described previously [26]. Only proteins and phosphosites with high localization probability (>0.75) with at least five individual SILAC ratios in each group were further used for statistical analysis. A two-sample unequal variance *t*-test was performed for the analysis of the PRE-R and PRE-NR groups while paired *t*-tests were performed to find significant differences in the PRE vs. 3D, PRE vs. 8D and 3D vs. 8D comparisons. Additional *Z*-statistics were carried out to calculate the significance of the fold change (FC) for protein expression and phosphorylation [38]. All the statistical analyses were carried out in Microsoft Excel. Pearson correlation coefficients (*R*) were calculated with Prism8 (GraphPad Software, San Diego, CA, USA). The hierarchical clustering of significantly differential proteins and phosphosites was performed with the Perseus platform by using the Spearman correlation function and average linkage. Standard fuzzy c-means clustering by the online VSClust application was carried to find distinct temporal profiles of protein expression and phosphorylation sites [39].

Gene ontology (GO) analyses were performed using a GO tool [40]. Prism8 was used to show the most significantly over-represented GO terms with *p* values <0.05. Venn diagrams were illustrated with the BioVenn online tool [41]. The amino acid distribution surrounding the phosphosites was analyzed using the iceLogo online tool [42]. Nonsignificant phosphosites were used as a reference set. A small number of phosphopeptide sequences were analyzed with the WebLogo website tool [43]. Kinase activity estimates were predicted with the online KSEA App. [44,45]. The whole phosphosite dataset was analyzed with the PhosphoSitePlus [46] and NetworKin [47] databases setting the substrate count and the NetworKin score cutoff to 5. Significantly regulated kinases with false discovery rate (FDR) <0.05 were shown in bar plots created with Prism8. Kinase prediction analysis by deep learning with the DeepPhos architecture was performed in the piNET web platform [48,49]. The kinase activation loop analysis was carried out with the online tool at http://phomics.jensenlab.org (accessed on October 2020). Protein–protein interaction (PPI) networks were obtained and visualized with STRING database (version 11.0) and the Cytoscape platform (version 3.8.0) as described elsewhere [26,50,51]. Reactome term enrichment was performed using the STRING app (1.6.0) [52]. Cellular signaling by regulated phosphoproteins were studied with the SIGnaling Network Open Resource (SIGNOR) 2.0 [53].

### 2.6. DNA Methylation Analysis

Patient leukemic cells were disrupted with TissueLyser (Qiagen, Germantown, MD, USA) used at 25 Hz for five minutes. Genomic DNA (gDNA) samples from 25 PRE-NR and 12 PRE-R patients were prepared with AllPrep DNA/RNA Mini Kit and QIAamp DNA Mini Kit (Qiagen). Five samples were further purified using the QIAamp DNA Micro Kit (Qiagen). Fifteen of the PRE-NR and nine of the PRE-R patient samples belong to the patient cohort used for MS-based proteomics analyses described in the present work.

Five-hundred nanograms (500 ng) gDNA was bisulfite converted using the EZ DNA Methylation^TM^ Kit (Zymo Research, Irvine, CA, USA) by the Genomics Core Facility of the Faculty of Medicine at the University of Oslo. The samples were further processed following Illumina’s Infinium HD Assay methylation protocol guide. Each sample was hybridized to an Illumina Infinium MethylationEPIC BeadChip and scanned using the Illumina iScan system. All analyses were performed by the Bioinformatics Core Facility of the same Faculty using R version 4.0.2. The analytical approach essentially followed selected steps described somewhere else [54]. Default values were used for all functions if nothing else was specified. The array sizes were different for some of the samples and therefore the force option was enabled when reading the data into R with the minfi package. This means that some of the samples could have had a reduction in the number of probes, to ensure that all samples had the same number of probes at the start of the analysis. The number of probes that were read into R from the .idat files was 865,859, and the Illumina manifest contained 867,530 probes. Quality control was performed and none of the samples in the dataset had a detection *p* value above 0.01, hence none of them were removed before analysis. The dataset was divided according to the comparing patient groups. Two normalization methods were used to assess whether different end results were generated. These methods were quantile and functional normalization from the minfi R package. There were two sets of probes used for the analysis of differential methylated regions: one set where the XY chromosome-associated probes were removed and another set where they were kept. During the filtering step, the probes with a detection *p* value above 0.01 in one or more samples were removed. Then, the probes associated with single-nucleotide polymorphism (SNPs) and cross-hybridizing probes were removed with the DMRcate R package. The XY chromosome associated probes were also removed in this step, if relevant.

## 3. Results

### 3.1. AML Patients Included in the Study

This study included 28 elderly and/or unfit patients (14 men and 14 women; median age 74 years with range 48–86 years). All patients had non-promyelocytic variants of AML. These patients represent the subset of patients with high relative and/or absolute levels of circulating leukemic cells in peripheral blood included in two previous clinical studies (see Material and Methods). The present study included several patients with high-risk leukemia, i.e., AML relapse (seven patients), secondary AML (ten patients), complex karyotype (five patients) and/or tumor protein p53 (*TP53)* mutations (two patients) (Table 1).

There is no general agreement with regard to which response criteria should be used for patients receiving low-toxicity AML stabilizing treatment [4], and we therefore evaluated the patients in the two previous clinical studies with regard to the conventional AML criteria for remission induction [55] and the criteria for the treatment of patients with myelodysplastic syndromes (MDS) [56]. Most of the responders in the two clinical studies showed disease stabilization according to the MDS criteria lasting for at least seven weeks; only two patients achieved complete hematological remission. All responders included in the present study had disease stabilization, except patient R2 who achieved complete hematological remission that lasted for 61 days. Eleven of the present patients were classified as responders to the treatment according to the MDS criteria; the other 17 patients did not respond to the treatment (Figure 1).

The 11 responders had a median survival of 132 days (range 58–644 days) from the start of the treatment, whereas the 18 non-responders had a median survival of only 19 days (range 2–112 days, Mann–Whitney U test, *p* > 0.0001). However, a clinical response with increasing peripheral blood cell counts is always seen within 14–21 days of treatment [4,6], and this was also true for the responders included in the present study. Even though our present study shows that the ATRA alters the proteomic profiles of the AML cells already after two days of treatment and additional proteomic alterations are detected five days after addition of VP (see Section 3.7, Section 3.8, Section 3.9 and Section 3.10], it may be argued that non-responder patients with a survival shorter than 14–21 days did not have time to obtain a detectable response when using response criteria based on bone marrow and peripheral blood cell counts. For this reason, we also compared the survival of our responders with the eight non-responders that survived for at least 21 days; it should then be emphasized that none of these responders showed any significant increase in normal peripheral blood cell counts of shorter duration at any time point before they died. However, the responders had significantly longer survival also when non-responders with survival shorter than 21 days were excluded from the analysis (*p* = 0.0034).

All the patients had clear signs of disease progression (i.e., clinical deterioration together with increasing peripheral blood blast counts, increasing bone marrow blast counts and/or decreasing normal peripheral blood cell counts) when they died. None of the patients had VP or TP levels above their therapeutic levels or any other signs of treatment toxicity when they died, and it should be emphasized that none of the patients had any evidence for ATRA syndrome.

### 3.2. The Pre-Treatment AML Cell Proteome for Responder (PRE-R) and Non-Responder (PRE-NR) Patients

We studied the proteome profiles of patient cells derived from 11 responders (PRE-R samples) and 17 non-responders (PRE-NR samples). Our proteomic dataset comprised 6330 quantified proteins, of which 4492 had a quantitative value in at least five patients per group. We found 98 differentially expressed proteins, where 55 proteins were upregulated and 43 were downregulated in the PRE-R relative to the PRE-NR group (Appendix A). The clustering analysis of the 98 differential proteins did not show a clear separation of the patient groups (Figure 2a).

GO enrichment analysis showed that the myeloid cell activation involved in immune response and antimicrobial humoral response (Figure 2b, top plot) were more abundant processes in PRE-R patients. These terms include proteins such as myeloperoxidase (MPO), elastase, neutrophil expressed (ELANE), FGR proto-oncogene, Src family tyrosine kinase (FGR), serpin family B member 1 (SERPINB1), chitinase 3 like 1 (CHI3L1) andH2B clustered histone 21 (H2BC21). Several calcium-binding proteins such as copine 3 (CPNE3), grancalcin (GCA), S100 calcium binding protein A8 (S100A8) and anoctamin 6 (ANO6) were also identified in this group. Most of these proteins are primarily located in the extracellular space and in the cytoplasmic vesicle part. The systemic lupus erythematosus Kyoto Encyclopedia of Genes and Genomes (KEGG) pathway, which represents proteins involved in the production of IgG autoantibodies that are specific for self-antigens such as DNA or nuclear proteins, was also enriched in the PRE-R patient group. Identified proteins belonging to this pathway are ELANE, cathepsin G (CTSG), H2BC21, H2A.Z variant histone 2 (H2AZ2) and H2B clustered histone 14 (H2BC14). 

In contrast, levels of proteins involved in processes such as hematopoietic or lymphoid organ development (e.g., sphingosine-1-phosphate lyase 1, SGPL1; SBDS ribosome maturation factor, SBDS), transcription by RNA polymerase II and cell death were higher in PRE-NR patients (Figure 2b, bottom plot; Table 2). Nearly 70% of the proteins upregulated in this group were annotated to intracellular membrane-bounded organelle in the cellular compartment analysis. The sphingolipid metabolism was the most significant KEGG pathway in this group with higher levels of SGPL1 and arylsulfatase A (ARSA).

The analysis of Reactome pathways showed that neutrophil degranulation (Reactome pathways I; Appendix A), M phase (Reactome pathways II) and the interconversion of nucleotide di- and triphosphates (Reactome pathways III) pathways were enriched in the set of 98 proteins regulated before treatment (Figure 2c). We observed two protein clusters mainly upregulated in the PRE-R patient group in Reactome pathways I. One of them was composed of 10 proteins mostly involved in catalytic antimicrobial activity and described in the GO and KEGG pathway enrichment analysis (Figure 2b, top plot) while the other was composed of four proteins connected to specific granule membrane (i.e., ANO6, phospholipase D1, PLD1; mast cell expressed membrane protein 1, MCEMP1; motile sperm domain containing 2, MOSPD2). A cluster of upregulated proteins in the PRE-R patient group consisting of several components of nucleosomes, i.e., histones, centromeres and the condensin complex, was found in the M phase Reactome pathway. The last Reactome pathway enrichment cluster, with different expression levels in PRE-R and PRE-NR patients, showed a significant enrichment of proteins involved in DNA synthesis and binding.

To summarize, pre-treatment AML cells derived from responders and non-responders showed complex biological differences, especially involving transcriptional regulation/RNA metabolism, and in addition, the responders showed increased levels of several molecules involved in degranulation/myeloid differentiation/intracellular transport and in cell cycle regulation.

### 3.3. The Pre-Treatment AML Cell Phosphoproteome for Responder (PRE-R) and Non-Responder (PRE-NR) Patients

A large dataset comprising 16,815 identified and quantified class I protein phosphorylation sites in 3501 proteins from the analysis of 11 PRE-R and 17 PRE-NR patient samples was elaborated in the Perseus platform. We found 107 differentially regulated phosphorylated sites from a starting 5519 phosphosites dataset that only included phosphosites quantified in at least five patients per group (Appendix A). Hierarchical clustering using these 107 phosphosites could not clearly distinguish samples from the two patient groups (Figure 3a). The differentially regulated phosphosite dataset comprised of 58 upregulated and 49 downregulated phosphosites in PRE-R relative to PRE-NR patients, respectively (Appendix A).

The cellular component GO analysis showed that cell cortex phosphoproteins were enriched in PRE-R patients, whereas the integral component of plasma membrane and extracellular matrix terms were enriched in the PRE-NR group (Figure 3b). Small molecule catabolic process and guanyl-nucleotide exchange factor activity were the biological process and molecular functional GO terms enriched in the PRE-R group (Table 3). In regard to the PRE-NR group, several phosphoproteins involved in mRNA splicing/binding (e.g., heterogeneous nuclear ribonucleoprotein M, HNRNPM; NPM1; microtubule associated protein 4, MAP4; matrin 3, MATR3) and in protein tyrosine kinase binding (interleukin-1 receptor accessory protein, IL1RAP; phosphoprotein membrane anchor with glycosphingolipid microdomains 1, PAG1) were significantly overrepresented.

The analysis of Reactome pathways identified two clusters of phosphoproteins associated to the metabolism of RNA (Reactome pathway I, Figure 3c). One of them involved RNA splicing factors such as serine/arginine repetitive matrix protein 2, SRRM2; splicing factor 3b subunit 1, SF3B1; splicing factor 1, SF1; and serine and arginine rich splicing factor 11, SRSF11; while the other one was composed of ribosomal protein S10, RPS10; ribosomal protein S6, RPS6; and RIO kinase 2, RIOK2. A sequence logo of the amino acid sequence windows surrounding the phosphorylation sites of the splicing factor network (Appendix A) identified the SP and the SXR/K motifs for mitogen-activated protein kinases (MAPKs) and protein kinase C delta/protein kinase cAMP-activated catalytic subunit alpha PRKCD/PRKACA, respectively, although RNA polymerase II subunit A (POLR2A) S1920 and SRSF11 S207 are phosphorylated by cyclin dependent kinase (CDK)1 and CDK2, respectively [46]. The phosphorylation of RPS6 at S236 by several members of the ribosomal protein S6 kinase family (i.e., RPS6KA1, RPS6KA3 or RPS6KB1) facilitates the assembly of the translation preinitiation complex [57] whereas polo like kinase 1 (PLK1) phosphorylation at RIOK2 S380 regulates mitotic progression [58]. Four phosphoproteins belonging to the apoptotic execution phase Reactome pathway II showed higher phosphorylation in PRE-R than in PRE-NR patients. The residue S18 is part of one of the five (S/T)P(K/A)K motifs found in the H1.5 linker histone, cluster member (H1-5) sequence which becomes phosphorylated, possibly by CDKs, during chromatin decondensation [59]. Clusters consisted of components of the histone deacetylation complex (i.e., nuclear receptor corepressor 1 and 2, NCOR1/2), and of RNA synthesis/elongation were observed in diseases of signal transduction and TP53 regulates transcription of DNA repair genes enriched Reactome pathways III and IV, respectively.

Further analysis of the regulated PRE-R vs. PRE-NR phosphoproteome by SIGNOR [53] identified five phosphoproteins involved in several signaling relationships (Appendix A). Among them, four phosphoproteins showed a higher phosphorylation in PRE-R patients (i.e., cofilin 1, CFL1; RPS6, protein tyrosine phosphatase non-receptor type 2, PTPN2; and MYC associated factor X, MAX). Only ATR interacting protein (ATRIP) showed a higher phosphorylation in PRE-NR patients.

The F-actin depolymerizing activity of CFL1 is inactivated by phosphorylation on S3 by NIK related kinase (NRK)/NIK-like embryo-specific kinase (NESK), testis associated actin remodeling kinase 1 (TESK1), protein kinase D1 (PRKD1) and LIM domain kinase 1/2 (LIMK1/LIMK2) [46,60,61]. Interestingly, some of these kinases were predicted to be activated in the PRE-R group by kinase–substrate enrichment analysis (KSEA) (Figure 3e, see results below). MAX forms a sequence-specific DNA-binding protein complex with myc proto-oncogene, bHLH transcription factor (MYC) and it can also repress MYC transcriptional activity from E-box elements [62]. ATRIP, which is more phosphorylated at S391 in PRE-NR patients, is an important component of the DNA damage checkpoint [63]. However, the role of that phosphorylated residue in its interaction with ATR serine/threonine kinase (ATR) is unknown.

A minor subset of exceptional five non-responders (also an extra one in Figure 3a) clustered outside the main non-responder cluster (left main cluster in both proteomic and phosphoproteomic analyses, see Figure 2a and Figure 3a) and among the responder patients. The only common characteristic of these six patients is that they were all *NPM1* wild type (wt), whereas the six patients with *NPM1* mutations were all included among the other 11 non-responder patients that clustered together in both analyses. This distribution of *NPM1* mutations in these two non-responder subsets reached statistical significance (Fisher’s exact test, *p* = 0.0338).

To summarize, the results described above reflect the complex differences in the phosphoproteome between responder- and non-responder-derived cells. Differences in transcriptional regulation and RNA synthesis/elongation seem to be of particular importance, and this is similar to the proteomic results described in Section 3.2. However, we also detected differences in kinase activity/intracellular signaling as well as in the regulation of cell cycle progression and apoptosis.

### 3.4. Altered Protein Phosphorylation Levels Are Not Caused by Altered Protein Levels 

In order to determine whether the protein phosphorylation levels were due to changes in protein abundance or to kinase activation, we investigated the expression of 85 phosphoproteins to the 107 differentially regulated phosphosites. We noticed that 80% of the phosphoproteins were not significantly changed at the protein expression level. However, we spotted five proteins significantly regulated at both the protein and phosphosite level, including RNA-binding and cytoskeleton molecules (Appendix A). The phosphorylation levels correlated closely with their protein expression levels (*R* = 0.992). This demonstrated that most of the differentially regulated phosphorylation sites in the PRE-R vs. PRE-NR study could not be explained by protein expression changes, which suggested an increased kinase-specific phosphorylation in the AML proteome.

### 3.5. Differential Kinase Activity in Pre-Treatment AML Cells Derived from Responders (PRE-R) and Non-Responders (PRE-NR) 

To identify protein kinases differentially activated in the PRE-R and PRE-NR groups, we performed phosphorylation site motif analysis (IceLogo) [42]. We found basophilic RXXpS/pT and RPPS motifs characteristics of PRKCA/PRKCD and dual specificity tyrosine phosphorylation regulated kinase 2 (DYRK2), respectively, in PRE-R patient samples when compared to the PRE-NR group (Figure 3d). Furthermore, the KSEA [44,45], which is based on phosphorylation FCs to estimate kinase’s activity, confirmed the higher activity of these kinases together with LIMK1/2 and CDK2 in the PRE-R group (Figure 3e).

CDK2 appeared to phosphorylate a large number of phosphosites in this dataset according to the KSEA search (e.g., PTPN2 S304, POLR2A S1920; stathmin 1, STMN1 S25, SRSF11 S207 and lamin A/C, LMNA S392). Only two kinases, protein kinase cGMP-dependent 2 (PRKG2) and microtubule affinity regulating kinase 1 (MARK1) involved in the activation of the extracellular signal-regulated kinases in osteoblasts and cell polarity, respectively [64,65], were predicted to be more active in the PRE-NR group. The prediction of phosphorylation sites by a novel deep learning architecture based on multi-layer convolution neural networks (i.e., DeepPhos) installed in the piNET web platform [48,49] confirmed the higher number of activated kinases on substrates more phosphorylated in PRE-R patients when compared with the PRE-NR group (Appendix A) as previously observed in the KSEA analysis. Among them, CDK1/2, MAPK4 (displayed as ERK1 in the figure) and PRKCD were again predicted to be more activated in the PRE-R group.

The search of our phosphosite dataset by the kinase activation loop tool (see Materials and Methods) did not find any phosphorylated residue in the domain activation loops. However, we identified two kinases (in addition to the RIOK2 identified previously), CDK13 and pre-mRNA processing factor 4B (PRPF4B) more phosphorylated at S395 and S397 in PRE-NR and at S578 and S580 in PRE-R patients, respectively. Strikingly, CDK13 and PRPF4B are both involved in RNA splicing and PRPF4B can phosphorylate CDK13 at S383, although it is unknown whether it could also regulate phosphorylation on the other two phosphosites [46].

### 3.6. The DNA Methylation of the AML Genome Does Not Differ when Comparing Pre-Treatment AML Cell Samples from Responders (PRE-R) and Non-responders (PRE-NR)

The analysis of differentially methylated regions in the AML genome of 12 PRE-R and 25 PRE-NR patients was performed with the DMRcate R package, as described in Materials and Methods, and *M* values were used to perform the statistical analyses. Significant changes of methylated regions were not found between PRE-R and PRE-NR patients (Appendix A). None of the normalization methods gave any significant regions. Even if the FDR cutoff was increased to 0.1, there were still no significant regions returned. 

These results suggested that there were no major differences in the methylated condition of the AML genome between responders and non-responders before treatment.

### 3.7. The Effect of the Triple Combination on AML Proteomic Profiles in Responders; Modulation of Translation, Organophosphate Metabolism, Intracellular Signaling and Mitochondrial Function

ATRA has not been tried as monotherapy but rather in various pharmacological combinations in previous studies of non-promyelocytic AML [4,5,6,10,17]. For this reason, we did not analyze the effect of ATRA as an independent treatment (i.e., 3D vs. PRE samples) but instead we used a bioinformatical strategy to analyze the effects of ATRA as a part of the triple combination that we regard as a relevant biological context. This strategy was used in Section 3.7, Section 3.8, Section 3.9 and Section 3.10.

In order to identify the temporal proteomic profiling before treatment (PRE) and after two days of ATRA treatment (3D) followed by combined ATRA–VP–TP treatment for five additional days (8D), we subjected the responders dataset to fuzzy *c*-means clustering analysis using the online VSClust application (Appendix A) [39]. Clusters corresponding to eight different response patterns were identified (Figure 4a, left plot; see Appendix A for all the proteins associated with each cluster).

The responder group clusters with low protein expression at 8D (cluster 1, 2 and 4) were enriched with the organelle part, rRNA processing and respiratory electron transport chain GO terms whereas clusters with a high protein expression at 8D (cluster 3 and 6) were enriched with extracellular region part, nucleobase-containing small molecule metabolic process and phosphate-containing compound metabolic process (Figure 4a, right plot). Descending (cluster 5) and ascending (cluster 7) temporal proteomic profiles where enriched with chromatin binding/histone modification (e.g., HDAC2; CREB binding protein, CREBBP) and T cell receptor signaling pathway/proteasome complex (e.g., proteasome 26S subunit, non-ATPase 11, PSMD11; proteasome 20S subunit alpha 2, PSMA2) GO terms, respectively (Figure 4a, right plot).

Furthermore, the analysis of Reactome pathways with responder proteomic samples showed seven pathways significantly enriched in the three different temporal pair-wise comparisons (Figure 4b, Appendix A). While proteins such as ribosomal protein L13 (RPL13) and eukaryotic translation initiation factor 5 (EIF5) of the GTP hydrolysis and joining of the 60S ribosomal subunit pathway were upregulated at 3D when compared to the pre-treatment time point, protein clusters belonging to interferon gamma signaling, respiratory electron transport, translation, ion channel transport and complex I biogenesis pathways had the lowest expression values at 8D of the ATRA–VP–TP treatment.

### 3.8. The Effect of the Triple Combination on AML Cell Proteomic Profiles in Non-Responders; Modulation of DNA Strand Elongation, RNA Processing, Actin/Cytoskeleton and Cholesterol Metabolism

The results of the fuzzy *c*-means clustering analysis for the five non-responders (Appendix A) are shown in Figure 5a (left plots). Clusters with low protein expression in the non-responder proteome at 8D (cluster 1 and 2) were enriched with extracellular organelle (e.g., ras homolog family member A, RHOA; cell division cycle 42h, CDC42) and the regulation of actin cytoskeleton organization GO terms, whereas clusters with high protein expression at 8D (cluster 3 and 4) were enriched with nucleic acid binding (e.g., high mobility group nucleosomal binding domain 2, HMGN2; DNA methyltransferase 1, DNMT1; lysine demethylase 2A, KDM2A) (Figure 5a, right plot). Clusters with the highest protein expression at 3D (cluster 5 and 6) were enriched with a nuclear part and RNA splicing (e.g., HNRNPC, HNRNPL and HNRNPK; SF1, SF3A2 and SF3A3) GO terms, whereas clusters with the lowest protein expression at 3D, cluster 7 and 8, were enriched with hydrolase activity/hydrolyzing O-glycosyl compounds and clathrin complex, respectively, when compared to pre-treatment and 8D conditions.

The analysis of Reactome pathways with non-responder proteomic samples showed seven pathways significantly enriched in the three different temporal pair-wise comparisons (Figure 5b, Appendix A). Integrin subunit alpha M (ITGAM), integrin subunit beta 2 (ITGB2) and intercellular adhesion molecule 3 (ICAM3) belonging to the integrin cell surface interactions Reactome pathway were upregulated at 3D when compared to the 8D time point. A protein cluster belonging to the DNA strand elongation Reactome pathway and composed of six DNA replication licensing factors (i.e., minichromosome maintenance complex components, MCMs) showed the highest expression values at 8D of the ATRA–VP–TP treatment.

Thus, the non-responder group differed from the responders regarding the effects on translation/transcription/RNA metabolism but also with regard to the additional metabolic effects in the responders. Even though a clinical response (i.e., increased normal peripheral blood cell counts) to ATRA/VP-based treatment can often be detected after 14–21 days, our present observations suggest that proteomic profiling can be used for the earlier identification of responder patients.

### 3.9. The Effect of the Triple Combination on AML Cell Phosphoproteomic Profiles in Responders; RNA Processing, Actin/Cytoskeleton and GTPase/Intracellular Signaling

The results of the fuzzy *c*-means clustering analysis of the phosphoproteomic profiles for five responders are summarized in Figure 6a (left plots) (Appendix A). Clusters corresponding to ten different response patterns were identified in the responder group; two of them with low site-specific phosphorylation at 8D (cluster 1–4) were enriched with RNA processing, mRNA splicing, via spliceosome (e.g., SRRM1/2; RNA binding motif protein 15, RBM15; serine and arginine rich splicing factor 2, SRSF2) and nuclear speck (e.g., SRSF1; RB binding protein 6, ubiquitin ligase, RBBP6) GO terms whereas clusters with high site-specific phosphorylation at 8D (cluster 5–7) were enriched with Rho GTPase binding, cytoplasmic mRNA processing body assembly and actin filament binding (Figure 6a, right plot). Clusters with high and low site-specific phosphorylation at 3D were enriched with the negative regulation of vasculature development/RNA polymerase I core binding (e.g., PML nuclear body scaffold, PML; programmed cell death 4, PDCD4; nucleolar and coiled-body phosphoprotein 1, NOLC1) and cyclin/CDK positive transcription elongation (i.e., RB transcriptional corepressor 1, RB1; CDK12) GO terms, respectively.

The analysis of Reactome pathways with responder phosphoproteomic samples showed one pathway significantly enriched in the 8D vs. PRE temporal pair-wise comparison (Reactome pathway I metabolism of RNA; Figure 6b) with higher phosphorylation on EIF4G1 (eukaryotic translation initiation factor 4 gamma 1) S1210, on EIF4B (eukaryotic translation initiation factor 4B) S406 and S409, and FIP1L1 (factor interacting with PAPOLA and CPSF1) S500. Differentially regulated phosphorylation sites found in the different temporal pair-wise comparisons can be found in Appendix A. Interestingly, the phosphorylation on HMGN1 S7, a phosphoprotein that might alter the interaction between the DNA and the histone octamer maintaining transcribable genes in a unique chromatin conformation, was found lower at 8D when compared to the pre-treatment and 3D time points.

To identify protein kinases differentially activated in the temporal pair-wise comparisons we performed phosphorylation site motif analysis with WebLogo [43]. We found the basophilic R/KXpS/pT and the SP motif characteristics of PRKCA/PRKCD and MAPK3/1, respectively, in pre-treatment patient samples when compared to the 3D group (Figure 6c, left plot). Furthermore, the basophilic RRRSXpS/pT, the SP and the acidic SXD/ED/E motif characteristics of AKT serine/threonine kinase 1 (AKT1), MAPK3/1 and casein kinase 2 alpha 1 (CSNK2A1), respectively, were observed in the 8D when compared to the pre-treatment group (Figure 6c, middle plot). Signaling pathways involving the phosphorylation of AKT1 and MAPK3/1 have been recently described in an independent study when using VP in combination with interferon 1 alpha (IFNA1)-Le [66].

To briefly summarize the important observations from this part of our study, we observed again that the triple combination altered the phosphorylation of proteins, especially those involved in the regulation of translation/transcription/RNA metabolism, but additional effects were also observed for cytoskeletal/actin proteins and Rho GTPase binding.

### 3.10. The Effect of the Triple Combination on AML Cell Phosphoproteomic Profiles in Non-Responders; Modulation of Translation/Transcription/RNA Metabolism and G-Protein Signaling

The results of the fuzzy *c*-means clustering analysis for five non-responders are summarized in Figure 7a (left plots) (Appendix A). Clusters with an increasing temporal phosphorylation profiling (clusters 1 and 8), i.e., from PRE to 8D, in the non-responder phosphoproteome were enriched with CDK activity (e.g., CDK1, CDK2, CDK11B, CDK12) and RNA splicing (e.g., DEAH-box helicase 16, DHX16) whereas clusters with a decreasing temporal phosphorylation profiling (clusters 2 and 5) were enriched with G-protein coupled receptor signaling pathway (e.g., IQ motif containing GTPase activating protein 2, IGGAP2; Rho guanine nucleotide exchange factor 7, ARHGEF7) and RNA polymerase II transcription factor activity (e.g., high mobility group protein HMG-I/HMG-Y and HMGA1) GO terms (Figure 7a, right plot).

Clusters with the highest protein phosphorylation at 3D (cluster 3, 4 and 6) were enriched with RNA splicing, protein–DNA complex assembly (e.g., H1.4 linker histone, cluster member, H1-4; heterochromatin protein 1 binding protein 3, HP1BP3; remodeling and spacing factor 1, Rsf-1, RSF1) and the sites of the DNA damage (e.g., replication timing regulatory factor 1, RIF1) GO terms, whereas clusters with the lowest protein phosphorylation at 3D (clusters 7, 9 and 10) were enriched with a negative regulation of the cellular catabolic process (e.g., PML, dyskerin pseudouridine synthase 1, DKC1), nuclear chromosome, telomeric region and the negative regulation of T cell activation (e.g., CD74 molecule, CD74; sialophorin, SPN) when compared to pre-treatment and 8D conditions.

The analysis of Reactome pathways with non-responder phosphoproteomic samples showed seven pathways significantly enriched in the three different temporal pair-wise comparisons (Figure 7b, Appendix A). H1-4 T18 and LMNB1 S391 belonging to the formation of senescence-associated heterochromatin foci (SAFH) Reactome pathway were more phosphorylated at 3D when compared to the pre-treatment time point. Several phosphosites on proteins belonging to the metabolism of RNA Reactome pathway (e.g., EIF4G1 S1188; tankyrase 1 binding protein 1, TNKS1BP1, S1666) showed higher phosphorylation at 8D of the ATRA–VP–TP treatment when compared to the pre-treatment condition. The higher phosphorylation on the latter phosphosites was also observed when compared with the 3D data. Phosphorylation at EIF4G1 S1188 by PRKCA induces binding to MAPK interacting serine/threonine kinase 1 (MKNK1), an interaction that has been linked to malignant transformation [67].

The analysis of kinase substrates using the WebLogo tool found an enrichment of basophilic R/KXpS/pT and the SP motif characteristics of PRKCA/PRKCD and MAPK3/1, respectively, in the 3D-NR and 8D-NR patient samples when compared to the pre-treatment group (Figure 7c, left and middle plot). The acidic SXD/ED/E kinase motif of CSNK2A1 was clearly observed in the 8D-NR when compared to the 3D-NR group (Figure 7c, right plot).

Taken together, the non-responder group differed from the responders both with regard to the phosphorylation events on the actin/cytoskeleton but also with regard to transcriptional regulation and GTPase/G-protein coupled signaling. Our observations therefore suggest that not only proteomic profiling but possibly also phosphoproteomic profiling can be used for early the identification of responders to ATRA/VP-based antileukemic therapy.

## 4. Discussion

The present study included patients from two different clinical studies. Both studies were based on ATRA and VP administered according to the described guidelines and thereby combined at low-dose and low-toxicity levels [4,6]. For these reasons, we included responders and non-responders from both these clinical studies in our present study as we have also done in a previous report, where the systemic metabolic effects of such AML stabilizing treatments were investigated [16]. The design of the initial treatment differed between the two studies and this was done to allow for the examination of early single drug effects in addition to the examination of the long-term effects of the combined treatment. Furthermore, even though the biological effects of both ATRA and VP could be detected during the first week of treatment, clinically relevant responses were usually observed after 2–3 weeks of treatment [4,6]. This was the case for both clinical studies, and for this reason, we compared the responders/non-responders from both studies because minor differences in drug administration during the first days of treatment are in our opinion not decisive for the later clinical responses that are often observed after 14–21 days. 

There is no general agreement with regard to the response criteria for AML patients receiving leukemia-stabilizing therapy. In the present study, we used the generally accepted definition of complete hematological remission [55], but we used the MDS criteria for the improvement/stabilization of normal peripheral blood cell counts, except that we required only one and not two months’ duration of this improvement [56]. AML is usually much more aggressive than MDS and has a median survival of only two–three months for elderly patients not receiving AML-directed therapy [68,69,70]. For this reason, we regarded even a shorter than two months improvement of peripheral blood cell counts to be unexpected and therefore sufficient for classification as a responder.

Both the karyotype and molecular genetic abnormalities are important prognostic factors for AML relapse [1]. Furthermore, several studies have demonstrated that the biological characteristics of the overall AML cell population (i.e., both the leukemic stem cells and the more mature cell population) are associated with relapse risk and survival, and these included mRNA gene expression profiles, noncoding RNA profiles, epigenetic and metabolic regulation [26,71,72,73,74,75,76,77]. Moreover, no morphological signs of residual bone marrow disease 14 days after the start of induction treatment and complete remission were achieved after one induction cycle, which are also favorable prognostic factors with regard to the risk of later AML relapse, and both of these criteria refer to the chemosensitivity of the overall AML cell population. For these reasons, we regard the investigation of the overall AML cell populations in our present study to be relevant. Finally, the response criteria for leukemia-stabilizing treatment refer to the overall leukemic cell population, and when the intention of the treatment is only disease stabilization and not leukemic stem cell eradication, the effects on the overall AML cell population will be of particular importance.

Based on our clinical studies, we can only conclude that the ATRA/VP-based treatment can give a clinically relevant disease stabilization with increased peripheral blood cell counts [4,6]. In addition, we also observed a longer survival for our responder patients compared with the non-responders in the present study, as it was also observed in the previously reported clinical studies. However, we would emphasize that the difference in survival should be interpreted with great care as it is not supported by observations in randomized clinical studies. 

Significantly different FCs for proteins and phosphosites between the PRE-R and PRE-NR groups were found according to our two-sample unequal variance *t*-test followed by a *Z*-statistics approach to address the significance of a protein ratio. AML cells derived from responders and non-responders differed with regard to a neutrophil degranulation molecular network. ARSA, a lysosomal protein that shows an altered expression in many malignancies, was the only protein that showed increased levels in non-responders (Appendix A). All the other proteins were significantly increased in leukemic cells derived from responder patients. Most interacting proteins included several proteins mainly expressed in myeloid cells and/or known to be released from the azurophilic granules (i.e., specialized lysosomes) or being lysosomal proteins. Previous studies have also suggested that the differentiation and differentiation induction is important for the response to VP in AML [10,78,79]. However, our present observations with the increased expression of several proteins known to be expressed in various organs/tissues suggest that other mechanisms reflecting more fundamental cellular functions may also be reflected in this network and thereby be important for induction of responses. Several proteins involved in carcinogenesis or leukemogenesis are increased for clinical responders (ANO6, CHI3L1, CTSG, ELANE and FGR), suggesting that the sensitivity to ATRA/VP additionally depends on more general functions involved in cancer development. This last hypothesis is further supported by previous observations describing associations between some of these network members and prognosis/survival in certain malignancies [80,81,82,83]. Finally, immune-mediated mechanisms may also be important for chemosensitivity because ATRA/VP reduces the levels of circulating T cells [5] and members of the neutrophil degranulation network are regarded as a possible leukemia-associated antigen (proteinase 3, PRTN3) and as a regulator of antileukemic immune reactivity (major histocompatibility complex, class I, B; HLA-B). Thus, the increased expression in responders of neutrophil degranulation network members may reflect different mechanisms for sensitivity to ATRA/VP-based therapy.

The expression of proteins involved in the cell cycle M phase was found to be upregulated in the responders before the start of the treatments (i.e., centromere protein E and K, CENPE and CENPK; CDK1, non-SMC condensing I complex subunit G, NCAPG; and several histones; Figure 2c). Both ATRA and VP have the capability to arrest cell cycle and induce apoptosis [84,85]. Thus, it seems that AML patients with a high expression of cell cycle proteins might benefit of the ATRA/VP treatments.

A gene expression study of 28 responders and non-responders to AML-stabilizing treatment ATRA/VP identified subsets of differentially expressed genes involved in transcriptional regulation (i.e., DNA/RNA binding proteins, transcription factors and nucleases), protein degradation/activation/modulation and metabolism [30] (data available at the Gene Expression Omnibus data repository with accession code GSE106096). As 23 out of the 28 patient samples overlapped with the patient samples we used in the present study, we compared the set of regulated expressed genes with our set of regulated proteins before treatment. As expected, the number of overlapped genes/proteins was low, probably due to different transcriptional mechanisms. However, the regulation of ELANE, NME/NM23 nucleoside diphosphate kinase 3, NME3, and hydroxyesteroid 17-beta dehydrogenase 11, HSD17B11, by both -omics approaches, well represent the biological mechanisms that are affected in the two patient groups (Table A1).

Previous experimental studies have identified several factors that can be important for the sensitivity of AML cells to the antileukemic effects of ATRA, but none of these factors differed significantly between our responders and non-responders. First, the transcriptional regulator MN1 was associated with ATRA resistance in a previous clinical study [86]. Second, experimental studies suggest that the transcription factor MECOM is important for the effect of ATRA in AML cells [17]. The high expression of this transcription factor is associated with chemoresistance in human AML, and for adult AML (but possibly not pediatric AML), high expression seems to have an independent prognostic impact for patients receiving conventional intensive chemotherapy [87,88]. Third, experimental studies suggest that the expression of lysine acetyltransferase 2B (GCN5) and lysine demethylase 1A (LSD1) may be important for the antileukemic effect of ATRA [89], but it is not known whether this is true during the complex situation of clinical treatment. Finally, aldehyde dehydrogenases are important for both the synthesis of retinoids and the metabolism of reactive aldehydes. However, aldehyde dehydrogenase activity may be generally high among our patients because both high-risk chemoresistance and AML in the elderly are associated with high protein levels of aldehyde dehydrogenase 2 family member (ALDH2) [90]. To conclude, there may be different explanations as to why these factors did not differ between responders and non-responders in our present study, but it should be emphasized that these previous and our present differences between responders and non-responders reflect the same biological processes of RNA transcription/translation/RNA metabolism.

AML is a very heterogeneous disease. We included consecutive patients in our present study that show an expected heterogeneity (Table 1). It is also expected that patients only receiving AML-stabilizing treatment will have a relatively high median age as was observed for our present patients. Furthermore, none of our patients had a favorable karyotype and this is also expected because these abnormalities are less frequent among elderly patients. Despite this heterogeneity, most of our non-responder patients clustered together when analyzing the proteomic (Figure 2a) and phosphoproteomic profiles (Figure 3a), but a subset of six patients were not included in the main non-responder clusters. These exceptional patients did not have *NPM1* mutation, and this difference reached statistical significance. We would emphasize that this observation should be interpreted with great care because the patients are few and the difference reached only borderline significance, but previous studies also suggest that ATRA therapy is less effective in patients with *NPM1* wt [91]. Our present study suggests that the AML cell biology for a subset of *NPM1*-insertion (INS)/fms-related receptor tyrosine kinase 3-internal tandem duplications (*FLT3*-ITD) negative non-responders differs from other non-responders, including *NPM1*-INS positive non-responders. Thus, our study supports the hypothesis that *NPM1* status is important for ATRA susceptibility.

Our present study included relatively old patients, especially when compared with previous studies of ATRA effects in patients receiving intensive chemotherapy (reviewed in [17]). Age-dependent differences may thus contribute to differences between the present and these previous studies. These could be both age-dependent differences in the AML cells, age-dependent differences in AML supporting stromal cells, e.g., osteoblasts or mesenchymal stem cells and epigenetic alterations [90,92,93,94]. This may also include an altered metabolism of ATRA in the bone marrow microenvironment [95]. Finally, the deregulated nutritional sensing and mitochondrial dysfunction are among the hallmarks of aging [96], suggesting that the ATRA/VP-induced modulation of systemic metabolic regulation occurs in a different metabolic context in our present patient compared with studies in younger patients [16].

Recent studies have shown that DNA methylation and histone acetylation are important epigenetic regulatory systems that are closely interrelated and mechanically dependent on each other [97]. Histone deacetylase inhibitors and DNA methylation inhibitors also seem to have overlapping effects on gene expression [98]. For this reason, we compared the DNA methylation status for responders and non-responders to antileukemic chemotherapy based on the ATRA/VP combination before treatment. We did not detect any significant differences in DNA methylation between the two groups. Our observations suggest that the DNA methylation status is not decisive for the effect of our present strategy of epigenetic targeting in the treatment of human AML and DNA methylation profiling cannot be used to predict the response to ATRA/VP therapy. However, in our opinion, the present observations do not exclude the possibility of additive or synergistic effects when combining these two pharmacological strategies in the treatment of human AML [99,100].

The AML phosphoproteome analysis of responders and non-responders showed an enrichment of phosphoproteins involved in transcriptional activity, RNA binding and RNA processing. The higher phosphorylation observed on apoptosis-related H1-3, H1-5, spectrin alpha, non-erythrocytic 1 (SPTAN1) and apoptotic chromatin condensation induce 1 (ACIN1) in responders might favor subsequent apoptotic events induced by the ATRA/VP treatment. Our kinase prediction analyses identified an enrichment of LIMK1/K2 and CDK substrates in the responder group. The latter seemed to correlate with the higher expression of CDKs observed in the proteome analysis. LIMK1 expression was significantly correlated with shorter survival of AML patients along with *FLT3* mutations, lysine methyltransferase 2A (*KMT2A*) rearrangements and elevated homeobox D10 (*HOXD10*) gene expression [101]. Furthermore, it was observed a reciprocal regulation between LIM kinases and CDK6. Our present work observed a high activity of both kinase types (not reciprocity) although LIMK1/2 activity was enhanced in treatment-responders. As the LIMK1-mediated cofilin pathway appeared to modulate retinoid receptor functionality, the role of LIM kinases in the survival of patients treated with AML-stabilizing therapy requires further investigations [102].

We have previously compared the protein expression and phosphorylation profiles at the time of first diagnosis for patients with later relapse or long-term AML-free survival [26] and patients at the time of first diagnosis and later relapse after previous complete remission [71]. Both studies included comparisons of chemosensitive vs. chemoresistant leukemic cells/disease, i.e., long lasting remission vs. only initial remission with later relapse and first diagnosis with initial remission vs. later chemoresistant relapse. We recently compared the AML cell proteomic and phosphoproteomics profiles between low-risk patients with favorable and high-risk patients with adverse genetic abnormalities [90]. When contrasting these studies with the present study where we compared pre-treatment cells derived from responders (i.e., with chemosensitive disease) vs. non-responders (i.e., with chemoresistant disease), we observed a very limited overlap between differentially expressed proteins (Appendix A–c). This lack of overlap suggests that the mechanisms of resistance and thereby the biomarkers with prognostic impact depend on the treatment given and would thereby be expected to differ between these four studies. This explanation is also supported by our previous clinical studies where several patients with high-risk chemoresistant disease according to conventional prognostic criteria (e.g., relapsed AML, previous chronic myeloproliferative neoplasia, complex karyotype) were classified as responders to our ATRA/VP-based treatment regimen [4,6]. A dependency of the prognostic impact of a certain biomarker on the received treatment has also been described in other diseases, e.g., the impact of the BCR-ABL translocation in acute lymphoblastic leukemia (ALL) treated with kinase inhibitors [103] and especially the Burkit B cell variant of ALL that was originally regarded as a high-risk variant but was later classified as a favorable variant when new therapeutic strategies were introduced [104]. 

The upregulation of proteins associated with neutrophil degranulation has been previously identified in relapse-free AML patients at the time of first diagnosis [26]. The overlap of regulated proteins in relapse-free and PRE-R samples was scant and only one protein, S100A8, of the three overlapping proteins was involved in neutrophil degranulation (Appendix A). Recently, nuclear S100A4, another member of the S100 protein family, has been suggested to be essential for AML survival [105]. Moreover, several proteins (ANO6, PLD1, ARSA, MPO and CPNE3) of the neutrophil degranulation cluster upregulated in PRE-R patients, with the exception of ARSA, were also found to be regulated in the paired sample study with AML cells derived at both first diagnosis and first relapse time points and in the high-risk vs. low-risk comparison (Appendix A) [71]. Thus, although neutrophil degranulation appears to be important in AML survival and treatment response, different sets of immune factors appear to regulate the different prognostic aspects of AML.

The overlap of significantly regulated phosphorylation sites found in the relapse vs. relapse-free [26] and in the current PRE-R vs. PRE-NR studies consisted of only six phosphosites (Appendix A). Among them, zinc finger protein 22 (ZNF22) S42 and DnaJ heat shock protein family (Hsp40) memberC5 (DNAJC5) S10 showed the same phosphorylation direction in PRE-R and relapse-free patients. Five significantly regulated phosphorylation sites were quantified in both the high-risk vs. low-risk and in the current PRE-R vs. PRE-NR studies (Appendix A). Phosphosites of thymopoietin (TMPO) involved in the nuclear anchorage of RB1 showed a higher phosphorylation on S156, S159 and T160 in PRE-R and low-risk patients.

We analyzed the overall effects of the triple combination (i.e., ATRA PRE–3D, VP 3D–8D and TP 3D–8D) on the proteomic and phosphoproteomic profiles of AML cells derived from responders and non-responders (Figure 4, Figure 5, Figure 6 and Figure 7). Thus, the ATRA effects were only analyzed as a part of the triple combination therapy. These comparisons showed that our pharmacotherapy had complex effects on the AML cells, but some main conclusions could still be made. First, ATRA induced the composite proteomic and phosphoproteomic responses both for responders and non-responders during the first stage of the triple combination therapy (i.e., PRE–3D), but many of these effects were further modulated during the continued treatment (3D–8D) with the triple combination. These include a DNA strand elongation network of nine upregulated proteins (mostly MCM complex components) in non-responders in the 8D vs. PRE comparison and proteins involved in the metabolism/deadenylation of RNA with an increased phosphorylation in the 8D vs. PRE and 8D vs. 3D comparisons, respectively (Figure 5b; Figure 7b). These subsequent delineated effects may either be late effects of the continued ATRA treatment and/or be caused by the VP/TP treatment. Interestingly, a recent study has observed that MCM7 polymorphisms may be able to predict the prognosis of AML patients [106]. Moreover, alterations in genes encoding MCMs have recently been found in the genomic profiles of aneuploid AML samples [107]. Second, the altered regulation of biological processes involving the processes of transcription/translation in addition to the already described RNA metabolism seemed to be important both for responders and non-responders and could be detected at 3D and 8D time points of treatment. Third, the triple combination altered the expression of proteins involved in the metabolic regulation in responders and in the cytoskeleton function in non-responders. Fourth, the upregulation of immune system proteins in responders at day 8 of the treatment (Figure 4a) suggested an activation of the immune system that might endure until the end of the therapy. Finally, the most important additional effects on phosphoproteomic profiles also involved GTPase activity and G-protein signaling for responders and non-responders, respectively. The phosphorylation of actin/cytoskeletal proteins was additionally altered in responders. In our previous phosphoproteomics study on relapse-free and relapse patients CSNK2A1, CDKs and PRKCA/PRKCD were found to influence AML prognosis [26]. Herein, while PRKCA/PRKCD were predicted to be activated in responders and non-responders in earlier stages of the triple combination, CSNK2A1 seemed to be activated in non-responders at day 8 of the treatment (Figure 7c) suggesting a CSNK2A1-VP crosstalk that might influence cell cycle progression.

Hundreds of ATRA responsive human genes have been previously identified in an evaluation of published data [108]. These genes were searched against the gene name of the regulated proteins and phosphoproteins found in the present study when we compared the 3D vs. PRE datasets (Appendix A). Two proteins, S100A8 and CD44 molecule (CD44), were recognized as RAR targets in non-responder patients. There are several possible explanations for this observation. First, the protein concentration is not only determined by the transcriptional regulation level but also by post-transcriptional mechanisms. Second, studies in various experimental models have described differences in the profile of ATRA-regulated genes between various cell types [109,110,111]. Therefore, the transcriptional effect of ATRA will be influenced by the cellular context and may therefore be further modulated by the molecular/biological characteristics of AML cells. Third, the protein profiles of the AML cells during ATRA monotherapy will also be influenced by additional indirect effects of ATRA as it has been previously observed at metabolism level [16].

AML is a very aggressive disease and many patients (especially elderly and unfit patients) survive only for a few weeks if they do not receive effective antileukemic treatment [112]. However, a clinical response to ATRA/VP-based treatment is usually observed after 2–3 weeks. The early identification of responders to the treatment will therefore be important so that the non-responders (i.e., most of the patients) can receive alternative treatment as early as possible before they deteriorate due to the disease progression. Our present results suggest that proteomic/phosphoproteomic AML cell profiling during the first week of treatment can identify the responders/non-responders to ATRA/VP-based treatment. Late responses after 2–4 cycles (i.e., 6–12 weeks) of therapy are also common for patients receiving AML-stabilizing treatment based on hypomethylating agents, an alternative but more toxic AML stabilizing treatment. The early identification of responders is also important with this treatment, and in our opinion, serial proteomic/phosphoproteomic profiling should be investigated as a possible tool for the early identification of responder patients receiving ATRA in combination with decitabine [10].

## 5. Conclusions

The use of ATRA and HDAC inhibitors is considered for the treatment of non-promyelocytic variants of AML. The use of these strategies is supported by both experimental and clinical studies, but the results from clinical studies suggest that (i) these agents should be used in combination with other antileukemic agents and (ii) the treatment will then only be effective for selected subsets of patients. Although new retinoids/HDAC inhibitors are now available for cancer treatment, it will be especially important to characterize the antileukemic in vivo effect of ATRA and the HDAC inhibitor VP to better understand the biological effects of these therapeutic strategies. 

Previous studies demonstrated that both ATRA and VP alters the systemic metabolic regulation in AML patients. In the present results, we used proteomic and phosphoproteomic profiling to characterize the in vivo effects of ATRA and VP on human AML cells. Our results suggest that the most important effects of combined in vivo treatment with ATRA/VP is the altered regulation of transcription/translation/RNA metabolism; these effects differ between responders and non-responders to the treatment, and additional effects that also differ between the two patient groups involve metabolic regulation, immunity, cytoskeletal function, and the regulation of GTPase activity/G protein signaling.

## Figures and Tables

**Figure 1 cancers-13-02143-f001:**
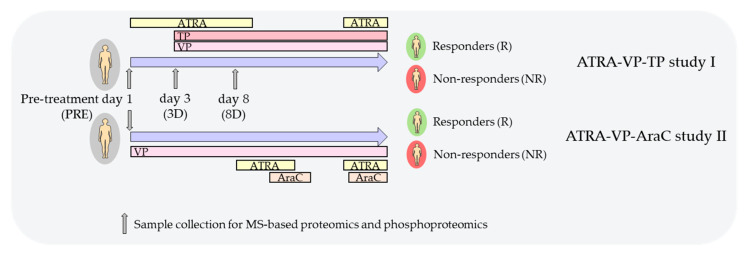
Overview of the two all-trans retinoic acid (ATRA)-valproic acid (VP)-based studies used in this manuscript [4,6]. Samples were collected before any treatment from both studies and were classified as responders (PRE-R) or non-responders (PRE-NR) according to the patient’s response at the end of the treatments. Samples from patients treated only with ATRA and with VP and theophylline (TP) afterwards were collected at day 3 (3D-R, 3D-NR) and at day 8 (8D-R, 8D-NR), respectively, in the ATRA–VP–TP study. All the time points for sample collection for further mass spectrometry (MS)-based proteomics and phosphoproteomics analyses are indicated with a grey arrow. The length of the lilac arrows does not correspond with the real treatments’ timing. The characteristics of both studies are detailed in Materials and Methods.

**Figure 2 cancers-13-02143-f002:**
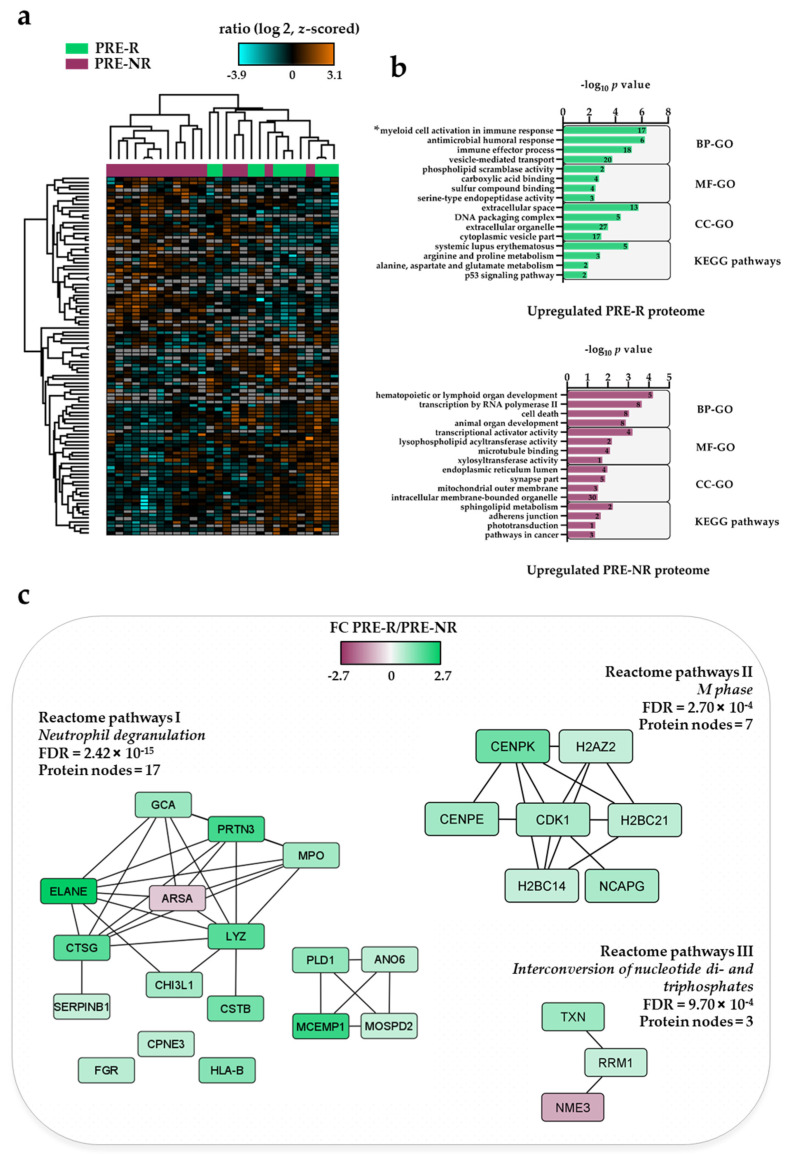
The acute myeloid leukemia (AML) cell proteome showed increased an abundance of proteins involved in neutrophil degranulation and in the M phase for responders before treatment: (**a**) the hierarchical clustering of 28 patient samples based on the expression (stable isotope labeling by amino acids in cell culture -SILAC- log_2_ ratio) of 98 proteins with significantly different regulation in AML cells from PRE-R (green) and PRE-NR (magenta) patients. Missing values are displayed in grey; (**b**) Gene Ontology (GO) and Kyoto Encyclopedia of Genes and Genomes (KEGG) pathways analyses of the proteins upregulated in PRE-R patients (top plot) and in PRE-NR patients (bottom plot) were performed to reveal enriched biological processes (BP; see the right side of the plots), molecular functions (MF), cellular compartments (CC) and KEGG pathways. GO terms and KEGG pathways are indicated on the y axis. The −log_10_ *p* value of the most significant GO terms and KEGG pathways are indicated on the x axis and the number of proteins corresponding to a GO term or KEGG pathway is shown at the end of each bar. * The full name of the GO term is myeloid cell activation involved in immune response; (**c**) Reactome term enrichment and false discovery rate (FDR) analyses were performed using the STRING app (1.6.0) in Cytoscape. The protein nodes are colored according to their PRE-R/PRE-NR fold change (FC), i.e., green indicates increased abundance in the PRE-R group and magenta increased abundance in the PRE-NR group.

**Figure 3 cancers-13-02143-f003:**
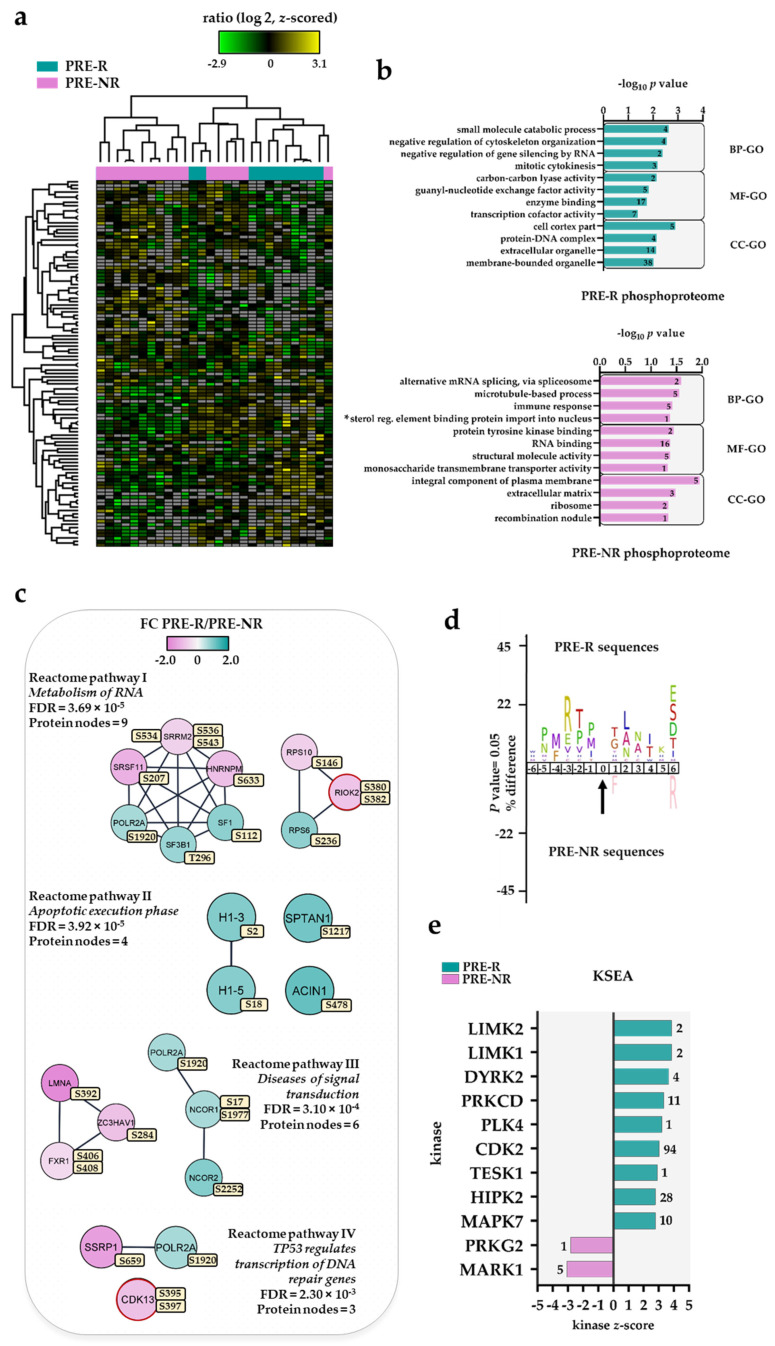
The pre-treatment AML cell phosphoproteome is enriched in RNA metabolism and apoptosis phosphoproteins, as well as cyclin dependent kinase (CDK) targets for PRE-R patients: (**a**) hierarchical clustering of 107 differentially regulated phosphorylation sites identified when 11 PRE-R (in green) and 17 PRE-NR (in pink) AML patients were compared. The SILAC log_2_ ratio scale and color code are also shown. Missing values are displayed in grey; (**b**) GO analyses of the phosphoproteins with higher phosphorylation in PRE-R (in green, top plot) and in PRE-NR (in pink, bottom plot) patients are shown as in Figure 2b. The number of phosphoproteins associated to a specific GO term or KEGG pathway are shown at the end of each bar. * The full name of the GO term is sterol regulatory element binding protein import into nucleus; (**c**) Reactome term enrichment was performed as described in Figure 2 and the phosphoprotein nodes are colored according to their PRE-R/PRE-NR phosphorylation FC, i.e., green shows increased phosphorylation in the PRE-R group and pink increased phosphorylation in the PRE-NR group. Nodes with red circles are illustrated for kinases; (**d**) kinase substrate analysis considering the ± six amino acids flanking the differentially regulated phosphorylation sites (see the black arrow) for each phosphosite cluster; (**e**) kinase–substrate enrichment analysis (KSEA) of the whole phosphoproteome dataset. The kinase *z*-score, on the x axis, represents the normalized score for each kinase (on the y axis), weighted by the number of identified substrates as shown at the end of each bar for PRE-R patients (in green) and PRE-NR patients (in pink).

**Figure 4 cancers-13-02143-f004:**
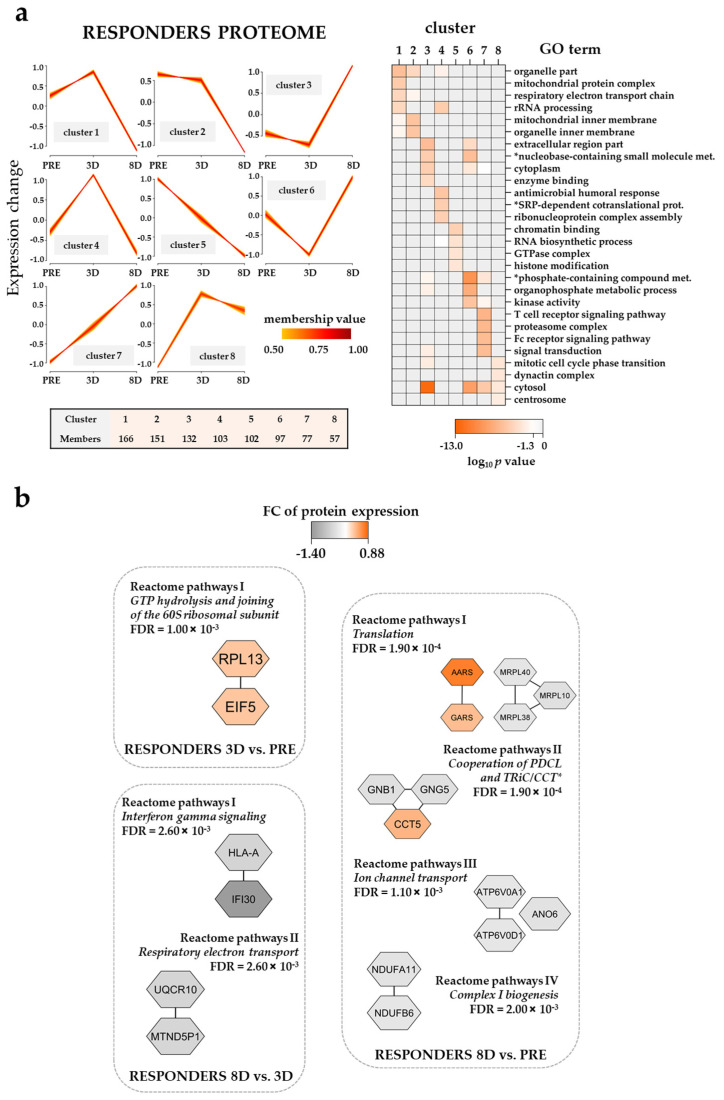
Protein expression profiles of responders before treatment (PRE) and after two days of ATRA treatment (3D) followed by ATRA–VP–TP treatment for five additional days (8D): (**a**) standard fuzzy c-means clustering of the protein expression profiles were performed by the online VSClust application (see Materials and Methods). The plots for the eight proteome clusters, adapted from the original VSClust plots, show only features confidently assigned with a minimum membership value of 0.5. The number of members for each proteome cluster is shown below the plots. The GO term enrichment analysis of each of the proteome clusters is shown as a heatmap on the right side. * Full names of abbreviated GO terms are nucleobase-containing small molecule metabolic process, SRP-dependent cotranslational protein targeting to membrane, phosphate-containing compound metabolic process and telomere maintenance via semi-conservative replication; (**b**) Reactome term enrichment of the regulated proteins identified in the differently temporal pair-wise comparisons was performed using the STRING app (1.6.0) in Cytoscape. The protein nodes are colored according to the corresponding 3D/PRE, 8D/PRE and 8D/3D FCs.

**Figure 5 cancers-13-02143-f005:**
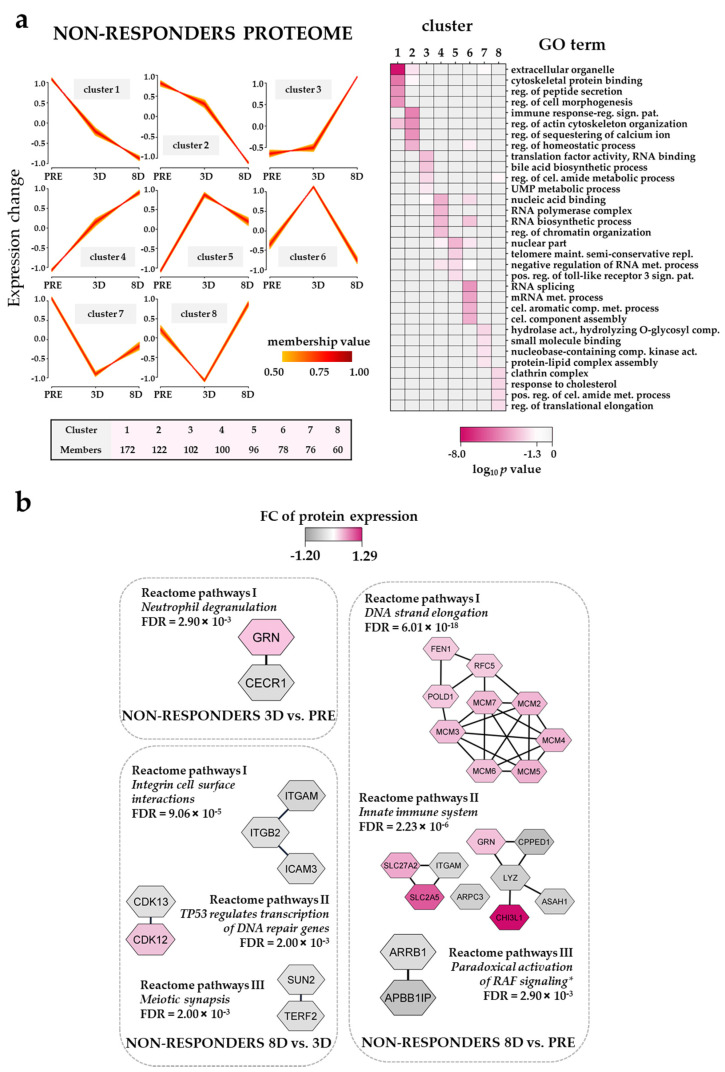
Protein expression profiles of non-responders before treatment (PRE) and after two days of ATRA treatment (3D) followed by ATRA–VP–TP treatment for five additional days (8D): (**a**) standard fuzzy c-means clustering of the protein expression profiles of non-responders at the three treatment time points and GO term enrichment analysis of each of the proteome clusters shown as a heatmap. Reg., sign., pat., cel., met., pos., comp., and act. were shortened from regulation, signaling, pathway, cellular, metabolic, positive, compound and activity, respectively, in the GO term nomenclature; (**b**) Reactome term enrichment of the regulated proteins identified in the differently temporal pair-wise comparisons were performed as indicated in the legend of Figure 4. * The full name of Reactome pathway III is paradoxical activation of RAF signaling by kinase inactive BRAF. The protein nodes are colored according to the corresponding 3D/PRE, 8D/PRE and 8D/3D FCs.

**Figure 6 cancers-13-02143-f006:**
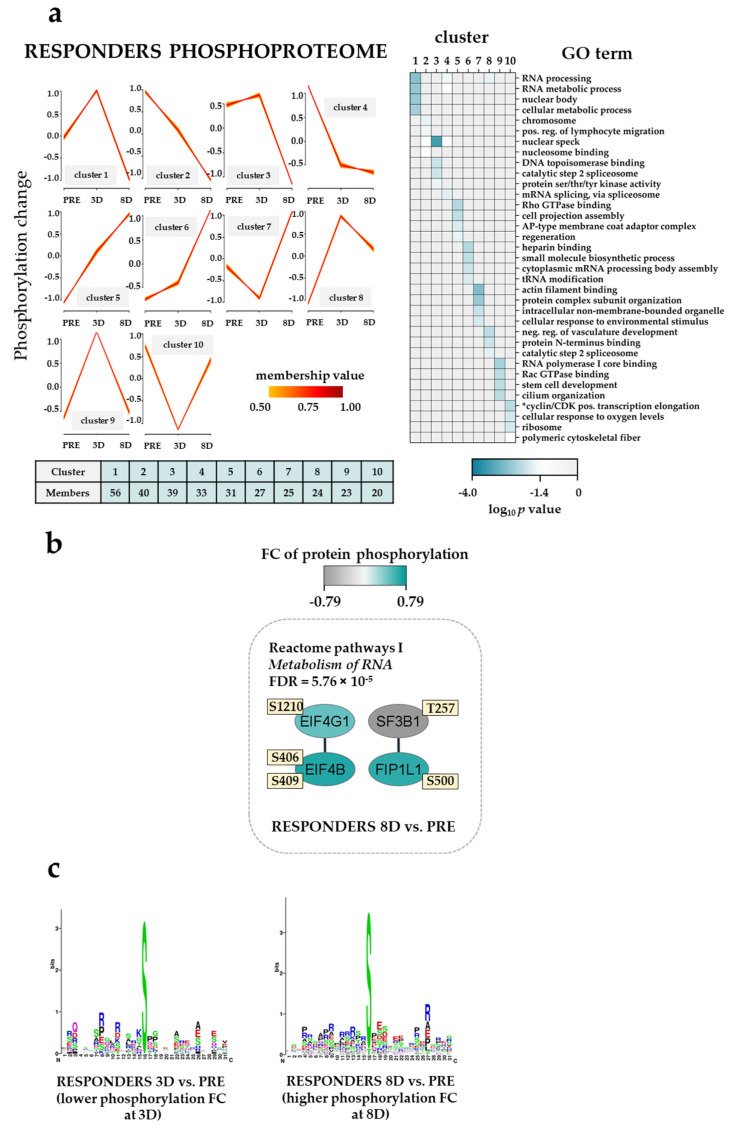
Protein phosphorylation profiles of responders before treatment (PRE) and after two days of ATRA treatment (3D) followed by ATRA–VP–TP treatment for five additional days (8D): (**a**) standard fuzzy c-means clustering of the protein phosphorylation profiles of responders at the three treatment time points and GO term enrichment analysis of each of the phosphoproteome clusters shown as a heatmap. * Full name of the abbreviated GO term is cyclin/CDK positive transcription elongation factor complex. Reg., pos. and neg. are abbreviations for regulation, positive and negative, respectively; (**b**) Reactome term enrichment of the regulated phosphoproteins identified in the temporal pair-wise comparison 8D vs. PRE which was performed as indicated in the legend of Figure 4; and (**c**) sequence motif analysis of the ± fifteen amino acids flanking the differentially regulated phosphorylation sites for the temporal pair-wise comparison 3D vs. PRE and 8D vs. PRE.

**Figure 7 cancers-13-02143-f007:**
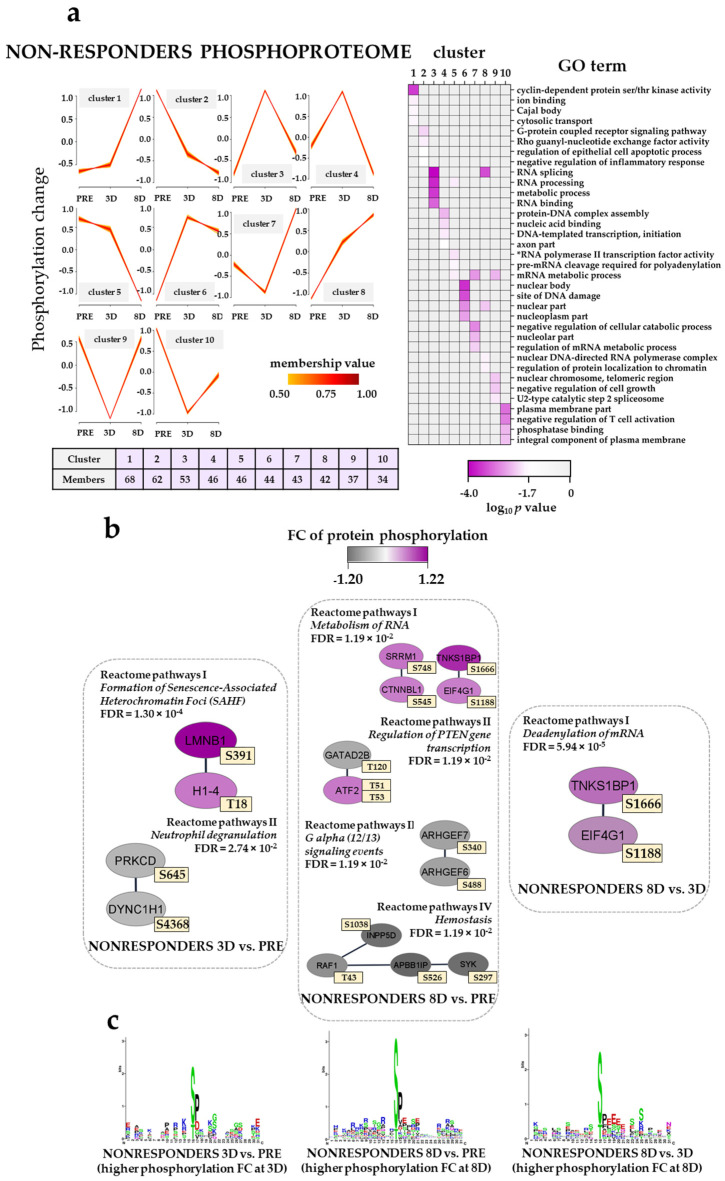
Protein phosphorylation profiles of non-responders before treatment (PRE) and after two days of ATRA treatment (3D) followed by ATRA–VP–TP treatment for five additional days (8D): (**a**) standard fuzzy c-means clustering of the protein phosphorylation profiles of non-responders at the three treatment time points and GO term enrichment analysis of each of the phosphoproteome clusters shown as a heatmap. * Full name of the abbreviated GO term is RNA polymerase II transcription factor activity, sequence-specific DNA binding; (**b**) Reactome term enrichment of the regulated phosphoproteins identified in the temporal pair-wise comparisons were performed as indicated in the legend of Figure 4; (**c**) sequence motif analysis of the ± fifteen amino acids flanking the differentially regulated phosphorylation sites for the temporal pair-wise comparison 3D vs. PRE, 8D vs. PRE and 8D vs. 3D.

**Table 1 cancers-13-02143-t001:** Clinical and biological characteristics of the included patients ^1^.

ID	Sex	Age	Previous Disease, Present Status	FAB	Membrane Molecule Expression ^2^					Karyotype	*FLT3*	*NPM1*	Additional Mutations	WBC Counts	Survival (Days) ^3^
					ANPEP	CD14	FUT4	CD33	CD34						
**Responders**															
R1	M	74	de novo	M0	+	-	-	+	+	multiple	wt	wt	*TP53*	18.7	151
R2	M	73	de novo	M1						nt	wt	INS	*IDH2, SRSF1*	12.1	383
R3 *	F	72	MDS	M2	+	-	-	+	-	*t* (1;5), *t* (2;3)	ITD	wt	*SETD2, RUNX1*	42.6	132
R4	M	81	polycytemia vera	M2	-	-	-	-	+	del (7)	wt	wt	*ASXL1, SRSF2, RAD21*	22.3	610
R5	F	77	MDS	M2	+	-	-	+	+	normal	wt	wt	*NRAS, TET2, ASXL1, RUNX1, SRSF1, STAG2, BCOR*	142.0	132
R6 *	M	80	de novo	M1	+	-	-	+	+	multiple	wt	wt	nt	8.0	58
R7 *	M	78	MDS	M1	+	-	-	+	+	nt	nt	nt	nt	142.0	69
R8 *	F	68	1st relapse	M1	+	-	+	+	+	normal	wt	wt	*TET2, ASXL1, BCOR*	15.6	105
R9 *	M	86	de novo	M4	+	+	+	+	-	nt	nt	nt	nt	18.7	59
R10 *	F	61	1st relapse	M1	+	-	+	+	+	multiple	wt	wt	*NRAS, SF3B1*	55.8	644
R11 *	M	62	2nd relapse	M2	+	-	+	+	+	del (7)	wt	wt	nt	4.9	350
**Non-Responders**															
NR1	F	82	de novo	M5	+	-	+	+	+	normal	ITD, TKD	wt	*WT1, DNMT3A*	142.0	37
NR2	F	60	relapse	M4	+	-	+	+	-	normal	ITD	INS	*DNMT3A, TET2*	16.7	12
NR3	F	77	de novo	M1	-	-	-	-	-	normal	ITD	INS	*DNMT3A*	68.5	32
NR4	F	78	de novo	M0	+	-	-	-	+	nt	wt	wt	*PTPN11, ASXL1, RUNX1, SRSF2*	21.0	18
NR5	F	82	polycytemia vera	M4	+	-	-	+	-	der (18); trisomy 8	wt	wt	*JAK2, GATA2*	32.5	19
NR6	M	71	chemotherapy	M4	+	-	-	+	+	normal	wt	INS	*KRAS, DNMT3A, TET2*	104.0	2
NR7	M	48	relapse	M4	+	nt	-	+	+	normal	ITD, TKD	INS	*DNMT3A, IDH1*	30.4	8
NR8	F	86	de novo	M0	-	-	-	+	+	del (5q)	wt	wt	*GATA2*	249.0	17
NR9	M	68	MDS	M0	-	-	-	-	+	normal	wt	wt	*TET2, ASXL1, CEBPA, SRSF2, STAG2*	1.5	24
NR10	F	77	de novo	M2	+	-	-	+	-	normal	ITD	INS	*DNMT3A*	77.8	17
NR11	F	70	MDS	M2	nt	nt	+	+	+	del (12)	wt	wt	*NRAS, KRAS, PTPN11, ASXL1, STAG2*	81.0	21
NR12 *	F	70	chemotherapy	M4	+	-	+	+	-	normal	wt	INS	*NRAS, DNMT3A, IDH1*	73.7	7
NR13 *	M	60	2nd relapse	M4	+	-	+	+	+	normal	ITD	wt	*WT1*	66.0	6
NR14 *	M	67	1st relapse	M1	+	-	-	-	+	normal	TKD	wt	none	15.6	73
NR15 *	M	68	myelofibrosis	M1	+	-	-	+	+	normal	wt	wt	*KRAS*	34.3	56
NR16 *	F	53	Li Fraumeni	M0	+	-	-	+	-	multiple	wt	wt	*TP53*	16.2	28
NR17 *	M	74	de novo	M0	+	-	-	-	+	multiple	wt	wt	*IDH2*	13.3	112

^1^ Abbreviations: ID, identification; FAB, French–American–British; *FLT3*, fms related receptor tyrosine kinase 3; *NPM1*, nucleophosmin 1; WBC, white blood cell; MDS, myelodysplastic syndromes; INS, insertion; ITD, internal tandem duplications; TKD, tyrosine kinase domain; wt, wild type; nt, not tested. ^2^ The results are presented as the percentage of positive cells. ^3^ Survival is presented as the survival from the start of the treatment. * These patients were included in the study by Ryningen et al. [6] and the other patients were included in the study by Fredly et al. [4].

**Table 2 cancers-13-02143-t002:** Upregulated proteins in non-responder patients sampled before treatment (PRE-NR) involved in several biological processes.

Gene	Protein Name	BP-GO ^1^
*ERG*	ETS transcription factor ERG	Transcription by RNA polymerase II
*NCBP2*	Nuclear cap binding protein subunit 2	
*NFKB2*	Nuclear factor kappa B subunit 2	
*SMAD4*	SMAD family member 4	
*CBFB*	Core-binding factor subunit beta	
*INTS12*	Integrator complex subunit 12	
*POLB*	DNA polymerase beta	Cell death
*NME3*	NME/NM23 nucleoside diphosphate kinase 3	
*CASP8*	Caspase 8	
*BRAT1*	BRCA1 associated ATM activator 1	
*RMDN3*	Regulator of microtubule dynamics 3	
*VAPA*	VAMP associated protein A	
*ARRB1*	Arrestin beta 1	Both terms
*MEF2D*	Myocyte enhancer factor 2D	

^1^ BP-GO stands for biological process gene ontology terms.

**Table 3 cancers-13-02143-t003:** Proteins in responder patients with higher phosphorylation before treatment (PRE-R) involved in catabolism and GTPase activity.

Gene	Protein Name	BP/MF-GO ^1^
*ALDOA*	Aldolase, fructose-bisphosphate A	Small molecule catabolic process
*BCKDHA*	Branched chain keto acid dehydrogenase E1 subunit alpha	
*PGM2L1*	Phosphoglucomutase 2 like 1	
*TRERF1*	Transcriptional regulating factor 1	
*SPTBN1*	Spectrin beta, non-erythrocytic 1	Guanyl-nucleotide exchange factor activity
*AKAP13*	A-kinase anchoring protein 13	
*SPTAN1*	Spectrin alpha, non-erythrocytic 1	
*RASGRP2*	RAS guanyl releasing protein 2	
*FLCN*	Folliculin	

^1^ BP/MF-GO stands for biological process or molecular function gene ontology terms.

## Data Availability

The LC–MS/MS raw files and MaxQuant output files have been deposited to the ProteomeXchange consortium via the PRIDE partner repository [149,150] with dataset identifiers PXD024110 for the pre-treatment samples and with PXD024197 for the pre- and post-treatment samples of responders and non-responders.

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
