# Peer review of "Proteomic Studies of Primary Acute Myeloid Leukemia Cells Derived from Patients Before and during Disease-Stabilizing Treatment Based on All-Trans Retinoic Acid and Valproic Acid"

_cancers, 2021, doi:10.3390/cancers13092143_

Round 1
Reviewer 1 Report
Major points:
- A proper introduction into nuclear receptors, such as RAR and RXR, and chromatin modifiers, such as HDACs is missing
- The sentence " ATRA is a highly potent agonist of all three cytoplasmic arginine-tRNA ligases (RARS); higher concentrations are needed at least for activation of other retinoic acid RXR receptors but further ATRA metabolism leads to the formation of potent RXR agonists [11]" is wrong and give this reviewer major concerns, whether the authors have not read their manuscript or not understand retinoid signaling. ATRA binds to retinoid acid receptors RARA, RARB and RARG but have no relation to "cytoplasmic arginine-tRNA ligases". Moreover, ATRA-related molecules, such as 9-cis retinoid acid, act as ligands of retinoid X receptors (RXRs).
- The bioinformatic analysis of the methylome data needs to be presented in more detail.
- How did the authors correct for inter-individual variation in gene and protein expression? This needs to be explained in the manuscript.
- The low level of clustering suggests that this study rather describes some 20 n=1 studies than two clearly testing groups of responders and non-responders. Please provide arguments in the manuscript.
- Which of the described proteins is encoded by a known RAR target gene?
Minor points:
- Abbreviations should be defined once at their first use and then consistently used. There are numerous issues throughout the manuscript, please carefully check.
- Gene name abbreviations should always be in italic, this applies also to figures and tables.
- Please double-check (e.g. via GeneCards) correct gene and protein names using latest HUGO nomenclature.
Author Response
Reviewer 1
Major points:
- A proper introduction into nuclear receptors, such as RAR and RXR, and chromatin modifiers, such as HDACs is missing
- The sentence " ATRA is a highly potent agonist of all three cytoplasmic arginine-tRNA ligases (RARS); higher concentrations are needed at least for activation of other retinoic acid RXR receptors but further ATRA metabolism leads to the formation of potent RXR agonists [11]" is wrong and give this reviewer major concerns, whether the authors have not read their manuscript or not understand retinoid signaling. ATRA binds to retinoid acid receptors RARA, RARB and RARG but have no relation to "cytoplasmic arginine-tRNA ligases". Moreover, ATRA-related molecules, such as 9-cis retinoid acid, act as ligands of retinoid X receptors (RXRs).
- The bioinformatic analysis of the methylome data needs to be presented in more detail.
- How did the authors correct for inter-individual variation in gene and protein expression? This needs to be explained in the manuscript.
- The low level of clustering suggests that this study rather describes some 20 n=1 studies than two clearly testing groups of responders and non-responders. Please provide arguments in the manuscript.
- Which of the described proteins is encoded by a known RAR target gene?
Minor points:
- Abbreviations should be defined once at their first use and then consistently used. There are numerous issues throughout the manuscript, please carefully check.
- Gene name abbreviations should always be in italic, this applies also to figures and tables.
- Please double-check (e.g. via GeneCards) correct gene and protein names using latest HUGO nomenclature.
We thank reviewer 1 for these comments to improve our manuscript. We have addressed all the points as follows:
Major points:
- We have now included additional information about histone acetylation, the various histone deacetylases and more details about various histone deacetylase inhibitors, especially valproic acid. This is now included in the Introduction section (lines 93-105) and three new references are included: Lakshmaiah KC, Jacob LA, Aparna S, Lokanatha D, Saldanha SC. Epigenetic therapy of cancer with histone deacetylase inhibitors. J Cancer Res Ther. 2014 Jul-Sep;10(3):469-78. doi: 10.4103/0973-1482.137937. PMID: 25313724.Chateauvieux S, Morceau F, Dicato M, Diederich M. Molecular and therapeutic potential and toxicity of valproic acid. J Biomed Biotechnol. 2010;2010:479364. doi: 10.1155/2010/479364. Epub 2010 Jul 29. PMID: 20798865; PMCID: PMC2926634.Bradbury CA, Khanim FL, Hayden R, Bunce CM, White DA, Drayson MT, Craddock C, Turner BM. Histone deacetylases in acute myeloid leukaemia show a distinctive pattern of expression that changes selectively in response to deacetylase inhibitors. Leukemia. 2005 Oct;19(10):1751-9. doi: 10.1038/sj.leu.2403910. PMID: 16121216.The introduction to nuclear receptors is addressed at the next point together with the correction of our wrong statement.
- We agree that additional information was needed and this part of the Introduction section has therefore been rewritten. We agree that ATRA should be regarded as a RAR agonist and this is now clearly stated and explained (lines 74-84). The following references have now been included:Akanuma H, Qin XY, Nagano R, Win-Shwe TT, Imanishi S, Zaha H, Yoshinaga J, Fukuda T, Ohsako S, Sone H. Identification of Stage-Specific Gene Expression Signatures in Response to Retinoic Acid during the Neural Differentiation of Mouse Embryonic Stem Cells. Front Genet. 2012 Aug 7;3:141. doi: 10.3389/fgene.2012.00141. PMID: 22891073; PMCID: PMC3413097.Wang K, Chen S, Xie W, Wan YJ. Retinoids induce cytochrome P450 3A4 through RXR/VDR-mediated pathway. Biochem Pharmacol. 2008 Jun 1;75(11):2204-13. doi: 10.1016/j.bcp.2008.02.030. Epub 2008 Mar 6. PMID: 18400206; PMCID: PMC2742682. Bushue N, Wan YJ. Retinoid pathway and cancer therapeutics. Adv Drug Deliv Rev. 2010 Oct 30;62(13):1285-98. doi: 10.1016/j.addr.2010.07.003. Epub 2010 Aug 3. PMID: 20654663; PMCID: PMC2991380.Heyman RA, Mangelsdorf DJ, Dyck JA, Stein RB, Eichele G, Evans RM, Thaller C. 9-cis retinoic acid is a high affinity ligand for the retinoid X receptor. Cell. 1992 Jan 24;68(2):397-406. doi: 10.1016/0092-8674(92)90479-v. PMID: 1310260.
- The DNA methylation analysis is now more detailed in the Methods sections (lines 276-280; 288-292). We hope our bioinformatics pipeline is better understood now.
- We do not understand what the reviewer meant by correcting the inter-individual variation in gene and in protein expression. AML patients are heterogeneous, and by using the well-established and generally accepted routine diagnostics such as karyotyping, mutational analysis and immunophenotyping/morphology it is difficult to find two identical patients. A heterogeneity is thus expected when we include consecutive (line 185) and unselected patients, and one would therefore also expect a proteomic heterogeneity/variation. Despite the well-known heterogeneity we find common and different characteristics across the heterogeneity. And the heterogeneity is not caused by variation in the number of leukemic cells. Therefore, we think that this variation should not be corrected for. Regarding technical variation, we work with highly enriched AML cell populations (>95%; line 183), follow a robust sample preparation as we have shown in previous papers (lines 192-197), do SILAC-quantification and perform statistical test for group differences and fold change of protein expression and phosphorylation (lines 230-236; 859-861).mRNA samples were obtained from the same cryopreserved batch and handled according to the highly standardized procedures. RNA quality control was performed in all the samples. As proteomic samples, all the mRNA samples were prepared in batches after a random selection. For inter-individual variation in the microarray analysis, quality control by principal component analysis was used to assess possible batch effects in the data. However, none was detected. We have used methodological approaches that avoid bias with regard to methodological variation, our methodology thereby gives a reliable and true picture of the patient heterogeneity.
- We have not included the details of the RNA sample preparation for not increasing the length of the manuscript but all those details can be found in the corresponding reference (line 898).
- AML is a very heterogeneous disease. In the present study we included consecutive patients and the heterogeneity of our patients is therefore expected. However, we did not have patients with favorable karyotype in our present study, and this is due to the fact that these abnormalities are much less frequent among elderly patients, and our study included mainly elderly and unfit patients. The present heterogeneity is consistent with the heterogeneity described in previous publications:Arber DA. The 2016 WHO classification of acute myeloid leukemia: What the practicing clinician needs to know. Semin Hematol. 2019 Apr;56(2):90-95. doi: 10.1053/j.seminhematol.2018.08.002. Epub 2018 Aug 22. PMID: 30926096.Moarii M, Papaemmanuil E. Classification and risk assessment in AML: integrating cytogenetics and molecular profiling. Hematology Am Soc Hematol Educ Program. 2017 Dec 8;2017(1):37-44. doi: 10.1182/asheducation-2017.1.37. PMID: 29222235; PMCID: PMC6142605.Döhner H, Estey E, Grimwade D, Amadori S, Appelbaum FR, Büchner T, Dombret H, Ebert BL, Fenaux P, Larson RA, Levine RL, Lo-Coco F, Naoe T, Niederwieser D, Ossenkoppele GJ, Sanz M, Sierra J, Tallman MS, Tien HF, Wei AH, Löwenberg B, Bloomfield CD. Diagnosis and management of AML in adults: 2017 ELN recommendations from an international expert panel. Blood. 2017 Jan 26;129(4):424-447. doi: 10.1182/blood-2016-08-733196. Epub 2016 Nov 28. PMID: 27895058; PMCID: PMC5291965.Papaemmanuil E, Gerstung M, Bullinger L, Gaidzik VI, Paschka P, Roberts ND, Potter NE, Heuser M, Thol F, Bolli N, Gundem G, Van Loo P, Martincorena I, Ganly P, Mudie L, McLaren S, O'Meara S, Raine K, Jones DR, Teague JW, Butler AP, Greaves MF, Ganser A, Döhner K, Schlenk RF, Döhner H, Campbell PJ. Genomic Classification and Prognosis in Acute Myeloid Leukemia. N Engl J Med. 2016 Jun 9;374(23):2209-2221. doi: 10.1056/NEJMoa1516192. PMID: 27276561; PMCID: PMC4979995.Arber DA, Orazi A, Hasserjian R, Thiele J, Borowitz MJ, Le Beau MM, Bloomfield CD, Cazzola M, Vardiman JW. The 2016 revision to the World Health Organization classification of myeloid neoplasms and acute leukemia. Blood. 2016 May 19;127(20):2391-405. doi: 10.1182/blood-2016-03-643544. Epub 2016 Apr 11. PMID: 27069254.Our intention was to have a study that is representative in general for elderly and unfit AML patients who only receiving leukemia-stabilizing treatment; for this reason we included consecutive patients as described in the article (line 185) and therefore it was expected that we would have heterogeneous patients. We hope this strategy can be accepted.With regard to the clustering we have included a separate comment/analysis of the outlier non-responders in the Results section (lines 500-507), and this is commented in a brief new chapter in the Discussion section (lines 924-940).
- We have found two proteins as RAR targets (Balmer and Blomhoff). These are S100A8 and CD44. This comparison was made using the regulated proteins and phosphoproteins described in Table S3-S6 in the 3D vs PRE datasets. This comparison has been included in the Discussion (lines 1061-1073).Balmer JE, Blomhoff R. Gene expression regulation by retinoic acid. J Lipid Res. 2002 Nov;43(11):1773-808. doi: 10.1194/jlr.r100015-jlr200. PMID: 12401878.
Minor points:
Thank you very much for these minor points. It was indeed needed a major correction on those issues.
- This is now multiply corrected (e.g., FDR, SILAC, LC-MS/MS, KEGG, several protein names, etc.) throughout the main text and the supplemental document
- This is now corrected throughout the main text and the supplemental document, especially on texts and tables. Protein symbols on figures are not in italic as they are in protein nodes.
- This is now corrected throughout the main text and the supplemental document. It has been applied to texts, tables and some of the figures (Figure 2, 3, 4 and 7). Most of the changes applied to histone proteins in the figures.
Reviewer 2 Report
Dr. Hernandez-Valladares and Colleagues presented a study focused on proteomic analysis of leukemic cells isolated in adult patients affected by non-promyelocytic AML before and during treatment based on ATRA and VP.
The Authors presented a variety of results and proteic/proteomic evaluations both considering the expression levels of proteins, their phosphorylation status and functions.
In order to better understand the value and the meaning of the presented results, I have some comments/questions.
- Statistics concerns: the Authors did not described the statistical analysis in Materials and Methods section. How did they calculate the sample size? Moreover, at line 428 the authors report an R value, but no correlation/linear correlation test has been described in Materials and Methods. A detailed paragraph about statistical evaluation must be reported.
- The Authors did not mention any controls. Did they used in vitro models such as cells lines treated or untreated with ATRA and VP? or healthy controls to appreciate the normal basal levels of proteins and their phosphorylation without therapy?
- Some information are missing. e.g. a) Line 161-162 "High peripheral blood blasts levels". The term High is not a measure. Please, define the used cut-off. b) Line 232 "Few samples". Few is ambiguous. Please, quantify the number of the samples. c) Line 507 and 509 "Lower" and "Higher" compared to what? Did the Authors fixed a threshold?
- Lines 100 to 105. The sentence is quite long and confused. Please, rephrased it.
- Discussion. I suggest to add discussions and comments about manuscript PMID 31111215 and PMID 30480765. The results presented in these publications may improve the interpretation of the results described in the present manuscript.
I hope my comments will help the Authors in improving the quality of the manuscript.
Author Response
Reviewer 2
Dr. Hernandez-Valladares and Colleagues presented a study focused on proteomic analysis of leukemic cells isolated in adult patients affected by non-promyelocytic AML before and during treatment based on ATRA and VP.
The Authors presented a variety of results and proteic/proteomic evaluations both considering the expression levels of proteins, their phosphorylation status and functions.
In order to better understand the value and the meaning of the presented results, I have some comments/questions.
- Statistics concerns: the Authors did not described the statistical analysis in Materials and Methods section. How did they calculate the sample size? Moreover, at line 428 the authors report an R value, but no correlation/linear correlation test has been described in Materials and Methods. A detailed paragraph about statistical evaluation must be reported.
- The Authors did not mention any controls. Did they used in vitro models such as cells lines treated or untreated with ATRA and VP? or healthy controls to appreciate the normal basal levels of proteins and their phosphorylation without therapy?
- Some information are missing. e.g. a) Line 161-162 "High peripheral blood blasts levels". The term High is not a measure. Please, define the used cut-off. b) Line 232 "Few samples". Few is ambiguous. Please, quantify the number of the samples. c) Line 507 and 509 "Lower" and "Higher" compared to what? Did the Authors fixed a threshold?
- Lines 100 to 105. The sentence is quite long and confused. Please, rephrased it.
- Discussion. I suggest to add discussions and comments about manuscript PMID 31111215 and PMID 30480765. The results presented in these publications may improve the interpretation of the results described in the present manuscript.
I hope my comments will help the Authors in improving the quality of the manuscript.
We thank reviewer 2 for these comments to improve our manuscript. We have addressed all the points as follows:
Point 1:
We briefly described the statistical analysis, but we have now extended it in more details including the calculation of the correlation coefficient, see lines 232-238. The correlation coefficient R has now been included in Table S2.
The present study is based on two nonrandomized clinical Phase II studies; for both studies it was required that one should have at least one responder among the ten first included patients to continue inclusion. This criterion was fulfilled for both studies. We expected at least 30% responders in both studies based on the experience from previous studies investigating ATRA plus VP alone without any additional drugs. We included all suitable patients during a 2-years period for the first clinical study and for a three years period in the second study; the number of patients was thus defined by the planned time period for inclusion of consecutive patients. As described in the clinical studies very few eligible patients refused inclusion. The first study then included 24 patients and the second 36 patients, the total number being 60 patients. We thus expected 10 responders in each of the studies, and we could then expect approximately 10 responders with high peripheral blast cell counts allowing the preparation of enriched AML cells by our simple standardized methodological approach. Thus, the patient number for both studies was determined by the time period and not by sample size calculation. Additional information is now included in the Material and methods section (lines 140-142; lines 157-158).
Point 2:
The present study is based on two nonrandomized Phase II clinical studies and for this reason we did not have a control group. This is now clearly stated in the Material and methods section (line 135).
We did not include analysis of normal cells in our present study because our intention was to characterize the heterogeneity between AML patients, i.e., the differences between responders and non-responders. Our unpublished data comparing AML blasts with normal CD34 bone marrow cells in a clustering analysis showed that the normal cells clustered separately from AML cells, but these data are not included in the present study because we do not regard them as relevant for our comparison of AML blast populations. We hope this is acceptable.
Point 3:
Thanks for these remarks. We have now defined “high levels of blasts”. Please lines 186-187. “Few samples” is now rewritten as five samples (line 265). The “lower” and “higher” should be low and high at 8D. This is now corrected on lines 607 and 609. Same corrections for a wrong use of “higher” and “lower” can be seen on lines 663, 667, 697, 701, 704, 890 and 977.
Point 4:
We have rephrased the original lines 100-105 to “These two pharmacological agents may not represent the final solution for the clinical strategies of RARA and HDAC inhibition, respectively. However, detailed studies of the in vivo effects of the two agents on primary AML cells may help us to understand the mechanisms behind the antileukemic effects of these two therapeutic strategies. Such information will probably also be relevant if more potent rexinoids and/or alternative HDAC inhibitors become available for clinical use”, on lines 117-123.
Point 5:
Thank you very much for these papers. It is important to describe independent works that support our findings. These facts are now written on lines 730-732 and 1042-1044.
Reviewer 3 Report
The text by Maria Hernandez-Valladares et al. is very interesting for hematologists who treat elderly patients or for whom can't be treated according to the usual chemotherapy regimens. The objective of the study is the proteonomic analysis of leukemic blasts during therapy with atra and valproic acid with or in the absence of chemotherapeutic agents. The section of the materials and methods part is well articulated as well as the results part. In the part of the results, the section relating to the characteristics of the patients should be inserted (file S1) The section of the discussion is well articulated and discussed with all the results obtained together with evidence present in the literature. The authors report several differences among responders and nonresponders in both proteonomics and phophoproteomics, demonstrating how the most important effects of combined in vivo treatment with ATRA / VP is altered regulation of transcription and RNA metabolism; these effects differ between responders and non-responders. The bibliography section is updated.
Questions:
authors describe no severe toxicity at reported doses of these nonchemotherapeutic agents, however the survival in both responders and nonresponders was quite short. Could the authors report if all deaths were disease related?
Le median survival of patients vas longer for responders than for nonresponders and this data should be integrated into the text (despite the low number of patients).
Author Response
Reviewer 3
The text by Maria Hernandez-Valladares et al. is very interesting for hematologists who treat elderly patients or for whom can't be treated according to the usual chemotherapy regimens. The objective of the study is the proteonomic analysis of leukemic blasts during therapy with atra and valproic acid with or in the absence of chemotherapeutic agents. The section of the materials and methods part is well articulated as well as the results part. In the part of the results, the section relating to the characteristics of the patients should be inserted (file S1) The section of the discussion is well articulated and discussed with all the results obtained together with evidence present in the literature. The authors report several differences among responders and nonresponders in both proteonomics and phophoproteomics, demonstrating how the most important effects of combined in vivo treatment with ATRA / VP is altered regulation of transcription and RNA metabolism; these effects differ between responders and non-responders. The bibliography section is updated.
Questions:
authors describe no severe toxicity at reported doses of these nonchemotherapeutic agents, however the survival in both responders and nonresponders was quite short. Could the authors report if all deaths were disease related?
Le median survival of patients vas longer for responders than for nonresponders and this data should be integrated into the text (despite the low number of patients).
We thank reviewer 3 for these comments to improve our manuscript. We have addressed all the points as follows:
We have now added Table S1 with patient characteristics as Table 1 in section 3.1, lines 317-323, and have better illustrated the study I on Figure 1 (new figure on lines 326-327).
Reviewer’s questions are addressed on lines 337-358 and further discussed at the Discussion part (lines 852-858).
[MdCHV1]It should be Table S1
Round 2
Reviewer 1 Report
Overall the manuscript has improved. However, there are still a few points that give concerns:
- The authors did not comment on their severe mistake in understanding retinoid signaling (or not proofreading their manuscript). The revised text is a bit better but the formulation is still misleading and partly incorrect. ATRA is without any doubt know since some 30 years as high affinity RAR ligands. Moreover, since nearly the same time 9-cis RA is an established RAR ligand. The authors should better cite these original findings in top journals rather than providing to the reader the impression that this issue is not settled. Additional findings about alternative RXR ligands may be cited, if this is necessary for the argumentation of the story.
- A retinoid concentration of 10-2 M (line 81), i.e. 10 mM is close to the solubility of the compounds and for sure not used in any experimental setting with cells or proteins.
- The author did not provide any information where their genome-wide data are available to the public. This is essential!
- Sirtuins are abbreviated SIRT.
Author Response
- The authors did not comment on their severe mistake in understanding retinoid signaling (or not proofreading their manuscript). The revised text is a bit better but the formulation is still misleading and partly incorrect. ATRA is without any doubt know since some 30 years as high affinity RAR ligands. Moreover, since nearly the same time 9-cis RA is an established RAR ligand. The authors should better cite these original findings in top journals rather than providing to the reader the impression that this issue is not settled. Additional findings about alternative RXR ligands may be cited, if this is necessary for the argumentation of the story.
- A retinoid concentration of 10-2 M (line 81), i.e. 10 mM is close to the solubility of the compounds and for sure not used in any experimental setting with cells or proteins.
- The author did not provide any information where their genome-wide data are available to the public. This is essential!
- Sirtuins are abbreviated SIRT.
Dear reviewer 1,
Please kindly read our answers to your points:
- We apologize one more time for our severe mistake on retinoid signaling. It was due to a wrong expression/writing of ideas and not because of our misunderstanding of the biology involved. We have corrected one more time paragraph 2 in the Introduction (lines 76-87) and we have added the suggested original manuscripts on retinoic acid receptors (number 11 and 12). We have now clearly stated that ATRA is a high-affinity RAR agonist. We have added a comment on possible indirect effects through modulation of systemic levels of lipid metabolites. We hope our solutions can be accepted.
- Based on the first and this second comment by the reviewer 1 the previous sentence/comment about ATRA concentrations has been left out (lines 77-82).
- The genome-wide data is available at Gene Expression Omnibus data repository with accession GSE106096. This is stated on lines 861-862
- This is now corrected on line 93
Kind regards and thanks
Reviewer 2 Report
The Authors addressed all my comments and concerns.
The manuscript has been significantly improved in term of quality and scientific soundness. Well done!
Author Response
Thanks a lot. We have improved the gramma at some sentences and remove some more typo errors. We hope this version represents now a clean one.